# Innate Receptor Activation Patterns Involving TLR and NLR Synergisms in COVID-19, ALI/ARDS and Sepsis Cytokine Storms: A Review and Model Making Novel Predictions and Therapeutic Suggestions

**DOI:** 10.3390/ijms22042108

**Published:** 2021-02-20

**Authors:** Robert Root-Bernstein

**Affiliations:** Department of Physiology, Michigan State University, East Lansing, MI 48824, USA; rootbern@msu.edu

**Keywords:** cytokine storm, cytokine release syndrome, hyperinflammation, innate immunity, synergy, COVID-19, SARS-CoV-2, Toll-like receptors (TLR), nucleotide-oligomer-recognition-domain-like receptors (NLR), bacterial coinfection, fungal coinfection, acute lung injury (ALI), acute respiratory distress syndrome (ARDS), sepsis, melatonin, TLR antagonists

## Abstract

Severe COVID-19 is characterized by a “cytokine storm”, the mechanism of which is not yet understood. I propose that cytokine storms result from synergistic interactions among Toll-like receptors (TLR) and nucleotide-binding oligomerization domain-like receptors (NLR) due to combined infections of SARS-CoV-2 with other microbes, mainly bacterial and fungal. This proposition is based on eight linked types of evidence and their logical connections. (1) Severe cases of COVID-19 differ from healthy controls and mild COVID-19 patients in exhibiting increased TLR4, TLR7, TLR9 and NLRP3 activity. (2) SARS-CoV-2 and related coronaviruses activate TLR3, TLR7, RIG1 and NLRP3. (3) SARS-CoV-2 cannot, therefore, account for the innate receptor activation pattern (IRAP) found in severe COVID-19 patients. (4) Severe COVID-19 also differs from its mild form in being characterized by bacterial and fungal infections. (5) Respiratory bacterial and fungal infections activate TLR2, TLR4, TLR9 and NLRP3. (6) A combination of SARS-CoV-2 with bacterial/fungal coinfections accounts for the IRAP found in severe COVID-19 and why it differs from mild cases. (7) Notably, TLR7 (viral) and TLR4 (bacterial/fungal) synergize, TLR9 and TLR4 (both bacterial/fungal) synergize and TLR2 and TLR4 (both bacterial/fungal) synergize with NLRP3 (viral and bacterial). (8) Thus, a SARS-CoV-2-bacterium/fungus coinfection produces synergistic innate activation, resulting in the hyperinflammation characteristic of a cytokine storm. Unique clinical, experimental and therapeutic predictions (such as why melatonin is effective in treating COVID-19) are discussed, and broader implications are outlined for understanding why other syndromes such as acute lung injury, acute respiratory distress syndrome and sepsis display varied cytokine storm symptoms.

## 1. Introduction: The Problem of What Causes Cytokine Overproduction Syndromes

Some patients experiencing severe COVID-19, the disease caused by the SARS-CoV-2 beta coronavirus, develop what is sometimes described as a “cytokine storm” or “cytokine release syndrome” characterized by the overstimulation of macrophages, dendritic cells and monocytes producing the cytokines interleukin 1 (IL1), interleukin 6 (IL6), interleukin 10 (IL10), tumor necrosis factor alpha (TNF-α), tumor necrosis factor beta (TNF-β) and ferritin [1,2,3,4,5,6]. These cytokines produce eosinopenia and lymphocytopenia characterized by low counts of eosinophils, CD8+ T cells, natural killer (NK) and naïve T-helper cells, simultaneously inducing naive B-cell activation, increased T-helper cell 17 (Th17) lymphocyte differentiation and the stimulation of monocyte and neutrophil recruitment [1,2,3,4,5,6]. Oddly, however, the release of cytokines interferon alpha (IFNα) and interferon gamma (IFNγ) are severely impaired in severe COVID-19 [1,2,3,4,5,6]. The result is a generalized, nonspecific hyperinflammatory response in the lungs resulting in acute respiratory distress syndrome (ARDS) with the concomitant activation of nonspecific inflammatory reactivity in the circulatory system and other organs, sometimes leading to multiorgan failure, leaky vasculature, coagulopathies and strokes [1,7]. Cytokine storms are, however, rare in COVID-19: a third of polymerase chain-reaction-positive SARS-CoV-2-infected people are symptom-free; most experience mild-to-moderate symptoms that are treatable on an out-patient basis, while only two to nine percent develop ARDS concomitant with clearly elevated cytokine levels that require hospitalization and intensive care unit treatment [8,9,10]. It is not known what triggers the nonspecific hyperinflammatory response in this minority of COVID-19 patients. Lacking a clear understanding of the mechanism or mechanisms at work, therapeutic approaches have had to develop on an ad hoc basis.

The problem of what causes “cytokine storms” or “cytokine release syndromes” extends beyond COVID-19 more generally to sepsis, acute lung injury (ALI) and acute respiratory distress syndrome (ARDS) associated with other respiratory infections [11,12,13,14,15,16,17,18,19]. All three are, like severe COVID-19, characterized by a release of unusual concentrations of cytokines and complicated by dysregulated and nonspecific hyperinflammatory responses that can lead to systemic complications [11,12,13]. Unfortunately, no agreed-upon definition of what constitutes a cytokine storm exists [14], complicating the search for clear diagnostic criteria and mechanisms to explain their causation. Thus, for example, IL-6 and interleukin 1RA (IL-1RA) increases were similar in some studies of influenza-associated ALI/ARDS patients and COVID-19 ARDS patients, but, as noted above, the interferon activity was significantly depressed in COVID-19 patients compared with influenza ARDS patients [15,16]. In other studies, while IL-6, IL-8 and TNF-α were significantly raised above their normal levels among COVID-19 ARDS patients, the increases were significantly less than in non-COVID ARDS or sepsis patients [17,18]. Whereas typical ARDS has an onset of less than one week after infection, COVID-19 ARDS averages eight to twelve days and tends to be less destructive to the lung epithelium [7]. Tumor necrosis factor-α (TNF-α) and interleukin 6 (IL-6) production by circulating monocytes is sustained in severe COVID-19 patients but not in bacterial sepsis or influenza-associated cytokine storms in which the monocyte numbers and function are impaired [16,19]. Overall, COVID-19 patients as a group (including mild and moderate cases) have ten to a hundred times lower cytokine levels than ALI/ARDS and sepsis patients [20]. Thus, not all COVID-19 patients experience cytokine storms and not all cytokine overproduction syndromes are the same. Accounting for who is susceptible to cytokine overproduction syndromes and their diverse characteristics is an important challenge for defining which COVID-19 patients are at risk and how to best treat them. Such understanding may also improve the diagnosis and treatments for ALI/ARDS and sepsis patients, as well.

The purpose of this paper is therefore to review the literature related to the innate immune receptors involved in activating cytokine release pathways and to develop a generalized model of how different viruses, bacteria and fungi associated with severe COVID-19, influenza-associated ALI/ARDS and sepsis affect these receptor systems. Particular emphasis will be placed on Toll-like receptors (TLR), nucleotide-oligomer domain-like receptors (NLR), NLR family pyrin domain containing proteins (NLRP) and retinoic acid inducible gene 1-like receptors (RIG or RLR) and their synergistic or antagonistic interactions when activated by single microbes and by polymicrobial infections. While a very robust literature exists on the innate receptor activation patterns (IRAP) of individual pathogens, this literature has not previously been integrated or compared in any systematic fashion, so as to develop a comprehensive model for analyzing such activation patterns. The resulting model demonstrates that severe COVID-19, influenza-associated ALI/ARDS and sepsis each induce different IRAP and that variations also exist even within these broader patterns that help to explain the diversity of cytokine storm or cytokine release syndrome variations. This model may also be useful for thinking about how best to prevent or treat such cytokine overproduction syndromes. 

## 2. Innate Immune System Receptor Activation in Cytokine Storms

### 2.1. The Presence of Multiple Concurrent Infections in Cytokine Release Syndromes

One assumption of the innate receptor activation pattern (IRAP) model is that multiple, concurrent infections explain the differences in observed cytokine storms that occur in severe COVID-19, influenza-associated ALI/ARDS and sepsis and also differentiate these cytokine-release syndromes from asymptomatic or mild COVID-19 and influenza virus or bacterial infections that do not lead to ALI/ARDS or sepsis. Influenza-virus-associated ALI/ARDS is found almost solely among patients with bacterial superinfections [21,22,23], so endotoxemia and antibodies against lipopolysaccharides (LPS), a bacterial cell wall breakdown product, are almost universally found in such ALI/ARDS patients [24]. Indeed, the symptoms of ALI/ARDS can be mimicked in rodent models merely by inoculating animals with LPS or exposing them to inhaled LPS (reviewed in [25]). Similarly, Wang et al. [5] noted that COVID-19-associated cytokine release is “exacerbated if combined with a superimposed bacterial infection”, while Sirivongrangson et al. [26] reported that, even though only a few COVID-19 patients had overt bacteremia, nearly all severe COVID-19 patients presenting with ARDS exhibited sepsis-like symptoms, the presence of bacterial 16S ribosomal RNA and evidence of bacterial endotoxins, both diagnostics for bacterial infection.

It is therefore noteworthy that severe COVID-19 accompanied by ARDS is distinguishable from mild COVID-19 by additional clinical parameter diagnostics for bacterial infection, including elevated ferritin [27,28], C-reactive protein [29,30,31] and procalcitonin levels [32,33], as well as eosinopenia [34,35]. Notably, IL-6 increases correlated strongly with CRP increases [29,30], suggesting a probable connection between cytokine overproduction and bacterial infection. Furthermore, cytokine-driven hyperinflammation in COVID-19 patient lungs is highly associated with the formation of neutrophil extracellular traps (NET) [36,37], which form readily in response to the presence of Gram-positive and Gram-negative bacteria and fungi [38,39] but are found only at low levels in viral infections [38]. Indeed, high rates of bacterial and fungal coinfections, especially with *Streptococcus pneumoniae, Haemophilus parainfluenzae, Klebsiella pneumoniae, Chlamydia pneumoniae, Mycoplasma pneumoniae, Aspergillus* and *Candida* species—more often than not diagnosed by polymerase chain reaction (PCR) and/or urinary antigen tests rather than isolation or culture—have been found to characterize severe COVID-19 patients and to distinguish them from mild and asymptomatic cases [40,41,42,43,44,45]. 

The roles of bacterial and fungal infections in cytokine storms and release syndromes will feature prominently in the model developed below. It will be argued that uncomplicated viral, fungal or bacterial infections do not activate sufficient innate immune system pathways to result in the hyperactivation required to produce an overproduction of cytokines. To achieve such hyperactivation, multiple synergistic innate systems must be co-activated, and that requires multiple, concurrent infections. Moreover, since different pathogens activate different sets of innate receptors, a plethora of virus–fungus–bacteria combinations are possible, each of which may manifest itself in a different form of cytokine overproduction.

### 2.2. Overview of Receptor Regulation of Cytokine Production

The cytokines characterizing cytokine storms or cytokine release syndromes are produced by the cells of the innate immune system (macrophages, dendritic cells and/or monocytes) upon exposure to pathogen-associated molecular pattern molecules (PAMPs) that characterize microbes and distinguish them from host antigens. Danger-associated molecular pattern molecules (DAMPs) such as fibrinogen and heat shock proteins that are released upon host cell death can also activate innate immune responses. PAMPs and DAMPs are recognized by diverse receptors on these innate cells, including Toll-like receptors (TLR), nucleotide-oligomer domain-like receptors (NLR), NLR family pyrin domain containing proteins (NLRP), retinoic acid inducible gene 1-like receptors (RIG or RLR) and melanoma differentiation-associated protein 5 (MDA5), among others. TLR and NLR can be grouped by whether they exist on the cell membrane (TLRs 1, 2, 4, 5 and 6); within cells on endosomal membranes (TLRs 3, 7, 8, 9 and 10) or in the cytoplasm (NOD1, NOD2, RIG1, MDA5 and NLRP3) [46]. Cell membrane-located receptors recognize mainly bacterial antigens, endosomal ones, mainly viral antigens and cytoplasmic ones, a mix of the two. The analysis that follows will focus on patterns of the microbial activation of human TLR, NOD-1, NOD-2 and NLRP3 among the NLR and RIG1 among the RLR, since these are the most commonly studied innate immune system receptors involved in sepsis, ALI, ARDS and COVID-19 research and, therefore, provide sufficient data for performing reasonable comparisons between these syndromes. Other innate receptors and their associated pathways undoubtedly also play important roles in cytokine overproduction that may nuance or modify the model presented below. 

Each receptor type is specific for limited classes of PAMPs and DAMPs. Among the most common PAMPs, TLR 1 recognizes triacyl lipopeptides; TLR2, diacyl lipopeptides and glycolipids; TLR3, double-stranded RNA and polyI:C, an artificial double-stranded RNA-like polymer used as an adjuvant; TLR4 recognizes lipopolysaccharides (LPS), heat shock proteins, fibrinogen and related proteins; TLR 5, flagellins; TLR6, diacyl lipoproteins; TLRs 7 and 8, single-stranded RNA; TLR9, viral and bacterial (CpG) DNA, which differs from eukaryotic DNA in having unmodified bases singularly characteristic of microbes and TLR10 recognizes retroviral proteins [47]. (Note that TLR10 has been omitted from the schematic diagrams below and further text discussions, because there is no evidence at present that retroviruses play a role in sepsis, ALI or ARDS in general or in COVID-19 in particular.)

Among the NLR, NOD1 recognizes γ-d-glutamyl-meso-diaminopimelic acid (i.e., DAP), a cell wall component of Gram-negative bacteria, while MDP recognizes N-acetyl muramyl-L-alanyl-d-isoglutamine (MDP), a component of both Gram-negative and Gram-positive bacteria. NOD2 can also be activated by single-stranded RNAs produced by various viruses such as respiratory syncytial virus and influenza viruses [47]. RIG1 recognizes viral RNAs and may be activated by the host cellular release of some microRNAs [47] Some host cellular DAMP can also stimulate NOD2, including Rho GTPases, which are activated by cells in response to infection, and by endoplasmic reticulum stress [48].

Among DAMPs, TLRs 1 and 2 recognize beta defensins; TLR2 also recognizes heat shock and surfactant proteins and high mobility group box 1 proteins and TLR4 recognizes the same proteins as TLR2, as well as hyaluronan, fibronectin and heparin sulfate. TLRs 7, 8 and 9 can recognize some host microRNAs and DNA fragments [49,50]. The DAMPs activating NLRP3 include cholesterol crystals and amyloid proteins such as those found in Alzheimer’s disease and diabetes [47]. Thus, once an innate immune response has been activated by the presence of foreign antigens, it may be maintained by the release of host antigens if the foreign antigens stimulate ongoing cellular damage or if an autoimmune disease process is initiated [51].

The types of cytokines released as a result of TLR, NLR and RIG1 activation depend on the sets of receptors activated by PAMPs and DAMPs. For example, TLRs 3, 4 and 6 can activate the Toll/IL-1 receptor domain-containing adaptor protein inducing INF-α activators (TRIF) pathway, resulting in interleukin production and a Th1 (or cellular) immune response. The other TLR along with TLR4 activate the myeloid differentiation primary response protein 88 (MyD88) pathway that results in the production of proinflammatory cytokines such as TNFα and IL6 and the production of a Th2 (antibody) response [52,53]. NLRP (including NLRP3) mediate the assembly of inflammasome complexes, leading to the activation of procaspase-1 and the release of ILβ and IL18, while RIG1, NOD1 and NOD2 mediate the assembly of complexes that mediate MyD88 activation via the mitogen-activated protein kinase (MAPK) and nuclear factor kappa-light-chain-enhancer of activated B cells (NF-κB) signaling pathways, resulting in the release of type 1 interferon, as well as the process of cellular autophagy [47,54,55]. These pathways, along with the main PAMPs and DAMPs that activate them, are summarized in Figure 1.

### 2.3. Synergistic and Antagonistic Receptor Interactions within the Innate Immune System

The release of cytokines is normal in any infection, so the problem of cytokine storms, cytokine release syndromes and systemic immune response syndrome reduces to the question of how to explain the *overproduction* of cytokines. Logically, the fact that the vast majority of individuals infected with SARS-CoV-2 or influenza A virus do not develop ARDS-related cytokine overproduction argues against any simple explanation, such as the virus itself being the sole necessary and sufficient trigger, which is a point that will be demonstrated evidentially in Section 2.7. Similarly, very few people infected with influenza viruses develop ALI/ARDS and its associated cytokine storm, again arguing against influenza viruses themselves as being sufficient causes of cytokine overproduction. The innate immune system has evolved to handle such uncomplicated infections by producing the well-regulated release of cytokines; thus, some additional mechanism must be at work.

An underappreciated aspect of innate immune function involving synergistic receptor interactions may be of paramount importance in resolving this problem. While any particular PAMP or DAMP activates only one or two innate receptor pathways, every innate receptor synergizes with some limited set of other receptors, so that combinations of PAMPs and DAMPs can initiate far greater releases of cytokines than individual ones. Figure 2A,B summarize the currently known sets of innate receptor synergisms based on two previous reviews [51,56] supplemented by additional sources that were found during the process of researching the present paper [57,58,59,60,61,62,63,64,65,66,67,68,69,70,71,72,73,74,75]. For example, TLR4 synergizes with multiple TLR, especially TLR2 and TLR9, as well as NLRP3 [51,56]. TLR2 coactivates TLR1 and TLR6, forming heterodimers and, also, synergizes with NLRP3 and NOD1. There is also TLR5-TLR7 synergy [57]; TLR5-TLR3 synergy [58,59]; NOD1 and 2 synergize with TLR2, 3, 4 and 9 [60]; RIG1 synergizes with TLR3 [61]; RIG1 synergizes with TLR3 [62,63]; NOD1/2 synergizes mostly with TLR [56,65]; TLR4 synergizes with NLRP3 but TLR2, TLR5 and TLR9 do not [70,71,72,73]; TLR2-NOD1 synergism [74] and NOD2-NLRP3 synergism [75].

Notably, the innate immune system is also characterized by receptor antagonisms to help regulate its function (Figure 2A,B). For example, RIG1 and TLR2 are antagonists; NOD2 pretreatment antagonizes TLRs2, 4, 5 and 9 [64]; NOD2 inhibits TLR1/2 [66,67], as well as TLR4, TLR9 and RIG1 [56]; TLR2, TLR4 and TLR5 inhibit TLR8 [68] and TLR9 inhibits TLR4 [69]. Sometimes, the same pair of receptors may, under one set of conditions, synergize with each other and, in another set of conditions, antagonize each other [51,56]. In the vast majority of cases, where both synergy and antagonism occur in the same pair of receptors, the synergy occurs when both receptors are activated simultaneously while the antagonism results when one receptor is activated significantly in advance (several hours to days) of the other. Thus, an agonist for one receptor may act as an antagonist for another [51,56]. These complications may explain some cases of conflicting data that will be discussed below with regard to whether particular sets of innate receptors are activated during cytokine storms of varying causes. 

It must also be cautioned that receptor synergisms and antagonisms are still being discovered and characterized, so as complex as the network illustrated in Figure 2A,B may be, it is likely to be incomplete, particularly with regards to antagonisms, which are less well-studied than synergisms. Additionally, as will be demonstrated explicitly in the following Section 2.7 and Section 2.8, the complexity of Figure 2A,B hides the fact that very few of these synergistic and antagonistic innate pathways are usually activated by any given infection, such as SARS-CoV-2, influenza virus, or any single bacterium or fungus. In practice, the sets of TLR and NLR that are activated or repressed are ordinarily quite small. On the other hand, as will be demonstrated in the next Section 2.6, severe COVID-19, influenza-related ALI/ARDS and sepsis are characterized by reasonably complex innate activity patterns (IRAP). These complex patterns are what need to be explained by whatever etiological agents trigger the cytokine storms that characterize these overproduction syndromes. 

### 2.4. A Hypothesis Concerning the Mechanism Producing Cytokine Storms

Let us pause, however, before proceeding to the analysis of IRAP in cytokine release syndromes and how they might be accounted for by first clearly stating the hypothesis that is to be tested and the alternative hypotheses against which it will be measured. The hypothesis to be explored here is that the cytokine overproduction in each case is caused by polymicrobial infections that activate multiple, synergistic innate receptors. Different sets of microbes exhibit different PAMPs and induce the expression of different DAMPs and, therefore, activate different sets of synergistic (and antagonistic) innate receptors, producing different distributions of cytokines. In consequence, cytokine storms or overproduction syndromes can exhibit multiple manifestations that, while similar in their systemic effects, are different in their causes and specific mechanisms. Due to these different mechanisms and manifestations, there is unlikely to be a single approach to treating cytokine storms and cytokine release syndromes that is effective for all. 

In order to test this hypothesis, the following sections will review what is known about the innate receptors activated in severe COVID-19, ALI/ARDS and sepsis to generate the IRAP characteristics of each type of overproduction syndrome. These “cytokine storm” IRAP will then be compared with the IRAP generated by the viruses associated with respiratory infections (e.g., SARS-CoV-2, influenza A virus, respiratory syncytial virus, adenoviruses, etc.), then by the bacteria and fungi most often associated with severed COVID-19 and ALI/ARDS (e.g., *Streptococcus pneumoniae, Haemophilus parainfluenzae, Klebsiella pneumoniae, Mycoplasma pneumoniae*, *Aspergillus, Candida*, etc.). If any of these individual pathogen IRAP can account for the IRAP of the cytokine release syndromes, then the hypothesis proposed here will be falsified, and a sufficient etiology will be established. On the other hand, if the proposed hypothesis is correct, then no single pathogen IRAP will be sufficient to account for the IRAP of the overproduction syndromes, so it will be necessary to investigate whether there are various combinations of viruses and bacteria and/or fungi that are sufficient.

In short, the innate receptor activation patterns (IRAP) of individual pathogens associated with severe COVID-19, ALI/ARDS and sepsis patients will be compared with the IRAP of these syndromes in order to evaluate the relative contributions that the individual or combined infections may make. These IRAP will then be evaluated using the model of synergisms and antagonisms summarized in Figure 2A,B to determine whether the sets of synergisms are sufficient to explain the cytokine storm or release syndromes.

### 2.5. Methods for Reviewing Literature Relevant for Comparing Alternative Hypotheses

Since the purpose of this paper is to review the literature relevant to testing the alternative hypotheses laid out in Section 2.4, the following method was used to choose and evaluate the relevance and usefulness of the sources. Eight types of literature were tapped that addressed the questions of what is known about: (1) innate receptor synergies and antagonisms, (2) the specific activation of innate receptors by individual microbes associated with COVID-19, (3) innate receptor activation in COVID-19, (4) the specific activation of innate receptors by individual microbes associated with influenza-associated ALI/ARDS, (5) innate receptor activation in influenza-associated ALI/ARDS patients, (6) the specific activation of innate receptors in sepsis, (7) the activation of innate receptors in sepsis and, finally, (8) the innate receptor antagonism by various treatments for cytokine release syndromes, especially focusing on COVID-19. Where possible, recent reviews of the relevant literature were employed, but where such reviews did not contain the needed information, recourse to PubMed searches for relevant studies was conducted and the most recent results utilized and, where possible, consensus results reported. For example, there are many good reviews of TLR–TLR synergisms that are cited in the next section but very few that address NLR–NLR or NLR–TLR synergisms, and antagonistic interactions are generally ignored in the review literature. Therefore, a systematic search was conducted on each pair of TLR and NLR in relationship to “antagonism” or “antagonist” or “synergism” or “synergy” (e.g., “NOD1 and TLR1 and synergy”) to try to capture any relevant studies. In some cases, relevant studies contradicted each other, in which case, a reference is made in the text below and in the various tables to the range of results obtained so as not to bias the discussion. Similarly, there are good reviews cited below that summarize what is known about the activation of TLR and/or NLR by some bacteria, fungi and viruses, but none covered the entire set of pathogens needed to test the alternative hypotheses put forward here, so, again, PubMed searches using relevant key terms such as “Klebsiella pneumoniae and NOD1” or “Mycoplasma pneumoniae and RIG1” were used to fill in the gaps as far as possible. In many cases, as will become apparent by the absence of entries in the tables below, no relevant studies could be located using this search procedure. The procedure was used, with similarly incomplete results, to attempt to capture everything that is known about TLR and NLR activation in COVID-19, ALI/ARDS and sepsis. In all cases, emphasis was put on acquiring the information from human studies (either clinical or laboratory ones utilizing human cells) rather than relying on animal models, and whenever the latter are used below, an explicit mention is made of this fact. No attempt at compiling a complete list of sources concerning cytokine storms of cytokine release syndromes or their treatments was made, but, rather, articles addressing key points of difference between the various hypotheses to be tested were selected and particular emphasis was put on finding studies that could provide data relevant to such tests. In sum, the primary criterion utilized in choosing sources to include in this review/hypothesis paper is whether a publication provided data that was useful to test some aspect relevant to differentiating the hypotheses and creating as complete a model of TLR-NLR activity in cytokine over-released syndromes as possible.

### 2.6. Synergistic and Antagonistic Receptor Activation Networks in Severe COVID-19, ALI/ARDS and Sepsis

Begin by considering what is known about innate receptor activation patterns (IRAP) in severe COVID-19, influenza-related ALI/ARDS and sepsis. Figure 2A,B can be used as a template for analyzing the probable sets of synergistic and antagonistic receptor interactions that can be expected in any given disease if the set of TLR, NOD, NLRP and RIG activities is known. Such data, as currently available from studies of human patients or experiments on human-derived macrophages, dendritic cells or monocytes (unless otherwise explicitly noted), are summarized in Table 1 for severe COVID-19 patients with ARDS, influenza-associated ALI/ARDS patients, sepsis patients and murine models of the latter two syndromes. Since all three syndromes are characterized by cytokine storms, comparing their activation profiles may illuminate the question of why the specific natures of these overproduction syndromes vary.

Innate immune receptor activation or inactivation has not been studied extensively for severe COVID-19-associated ARDS, but consistent evidence exists for the activation of TLR4 [76,77], TLR7 [78] and NLRP3 [79,80,81]. Conversely, increased levels of NR3C1, an NLRP3 antagonist, decreased the COVID-19 severity [82]. Conflicting reports for the activation of TLR2 [76,83], TLR3 [76,84] and TLR9 [76,77] also exist. Consistent data indicate that TLR1, TLR5, TLR6, TLR8, NOD1, NOD2 and RIG1 are not activated in severe COVID-19 [76,77]. These data distinguish COVID-19-associated ARDS from influenza virus-associated ALI-ARDS, which is characterized by the activation of TLR3, TLR4, TLR7 and NLRP3 and the downregulation of TLR2 and RIG1 [85,86,87,88,89,90]. Murine models of sepsis-induced ALI mimic TLR3, TLR4, TLR7 and NLRP3 activation but, unlike the human disease, also activate TLR2 and TLR9 [91,92,93,94]. Human sepsis patients have extremely diverse innate receptor activation profiles. Silva et al. [95] found no changes in the protein expression of TLR2, TLR4 or TLR9 and the upregulation of TLR5 in human sepsis patients, while Härter et al. [96] reported that TLR2 and TLR4 were the main receptors upregulated during sepsis, a finding confirmed by Gao et al. [97] and Kumar [98] in their patients, who also found increased expression of TLR3 and TLR7. Armstrong [99], meanwhile, found increased TLR2 mRNA and protein expression and increased TLR4 mRNA but no increase in protein, which was the exact opposite of the results reported by Brandl et al. [100]. RIG1 mRNA was also upregulated in human sepsis patients but not the protein expression [101]. In sum, all that can be said in general about innate activation in sepsis is that TLR2, TLR4, TLR5, TLR7 and NLRP3 [102] can be, but are not necessarily, activated, while TLR9 may or may not be downregulated [103,104,105]. The same generalizations can be made about murine polymicrobial sepsis models [91,106,107,108], suggesting that sepsis is not a single, definable disease [97]. Table 1 summarizes these data, demonstrating that cytokine release syndromes share only partial overlaps in their innate receptor activation profiles, which provides a possible clue as to why their cytokine release profiles also differ and, therefore, why it has been difficult to define a clear set of diagnostic criteria for them or to devise a comprehensive or universal approach to treatment.

Parenthetically, it is noteworthy that experiments utilizing mRNA expression as a measure of receptor production not only fail to mirror the results of direct measurements of protein expression or receptor activation but often yield contradictory results [99,100,109] (Table 2). The reasons for these results are obscure and beyond the scope of the current paper but may potentially be of significance in understanding the regulation of protein expression systems in the highly activated disease states being discussed here and may argue against using mRNA expression as a clinical measure. 

The data in Table 1 can be incorporated into the activation profile template provided in Figure 2A,B to elucidate the likely synergisms and antagonisms elicited by receptor activation. The result for severe COVID-19 is shown in Figure 3A. As Figure 3A illustrates, severe COVID-19 is characterized by seven innate receptor synergisms that may be offset by four antagonistic interactions (though, as noted above, antagonisms are often the result of the activation of one receptor significantly in advance of the other, which may, or may not, be the case in COVID-19). These synergisms, even moderated by some antagonistic interactions, help to explain the cytokine overproduction that characterizes severe COVID-19.

The figure for influenza-associated ALI/ARDS is shown in Figure 3B and is identical to that for COVID-19, except that TLR9 is not activated, so its synergism with TLR4 is absent, as is its antagonism of TLR4. Again, the significant number of innate receptor synergisms suggests the cause of cytokine overproduction in this syndrome, while the large number of antagonisms suggest why some TLR and NLR that might be expected to be activated are not. 

Finally, the IRAP for sepsis is illustrated in Figure 3C, which has very similar numbers of synergisms and antagonisms as ALI/ARDS but differs significantly in which TLR and NLR participate in these interactions. There are great uncertainties attending to the innate receptor activation in sepsis cases, so the synergy/antagonism profile provided here must be taken as tentative. Still, it should be obvious that if TLR2, TLR4, TLR7 and NLRP3 are all activated, as they appear to be in most human patients and in animal models such as murine polymicrobial sepsis (Table 1), then the sepsis profile will differ from that of severe COVID-19 and ALI/ARDS, since it is lacking activations, and attendant synergisms/antagonisms, involving TLR3 and TLR9, possibly substituting these with activation of TLR5 or TLR6. These differences might argue for an essential role of virus PAMPs in COVID-19 and influenza-associated ALI-ARDS, as would be expected, but no or a minor role for viral PAMPs in the etiologies of most cases of sepsis. So, once again, the pattern is such that it suggests the cause of the cytokine storm associated with sepsis but also helps to explain why such cytokine storms differ in their details from those associated with COVID-19 and ALI/ARDS. 

One message to take home from this section is that all types of cytokine release syndromes share the task of activating multiple TLR and NLR that results in multiple synergistic interactions well in excess of the number of antagonisms. The excess of multiple synergisms may account for the supranormal release of cytokines. The second message to take home from this section is that the specific sets of TLR and NLR activated in severe COVID-19, influenza-associated ALI/ARDS and sepsis differ in ways that may consequentially alter the specific nature and magnitude of cytokine releases.

### 2.7. Varied Receptor Activation by PAMP Produced by Different Pathogens

The previous Section 2.6 raises the question of how to explain the activation of the particular sets of innate immune system receptors that participate in the numerous synergistic and antagonistic interactions present in severe COVID-19, ALI/ARDS and sepsis. The hypothesis outlined in Section 2.4 proposes that the mechanism may involve multiple, concurrent infections. One way to test this proposition is by examining an alternative hypothesis, which is that individual causative agents suffice to activate the sets of receptor networks associated with COVID-9, ALI/ARDS and sepsis. Thus, perhaps SARS-CoV-2, influenza A virus and individual bacteria can each account for the innate receptor activation patterns found in their respective cytokine release syndromes. Therefore, the question becomes whether the activation profile in severe COVID-19 is simply a reflection of the effects of SARS-CoV-2 PAMP expression or are other factors (e.g., bacterial or fungal activation) needed to explain the profile? Similarly, is the innate activation profile of influenza-associated ALI-ARDS a result of the influenza A virus or coinfections with bacteria such as *Streptococcus pneumonia* or *Haemophilus influenzae*? Can sepsis profiles be explained by individual bacterial infections, or are they better explained by combinations of bacteria or bacteria working in conjunction with viruses or fungi? Comparing the innate activation profiles (IRAP) of the various viruses, bacteria and fungi associated with COVID-19, ALI/ARDS and sepsis permits these possibilities to be evaluated.

Table 2 summarizes the known TLR, NLR, NLRP3 and RIG1 activation patterns of several respiratory viruses: human coronavirus type 229 (CoV-229) (which causes cold symptoms), severe acute respiratory syndrome coronavirus type 1 (SARS-CoV-1), severe acute respiratory syndrome coronavirus type 2 (SARS-CoV-2), Middle East respiratory syndrome virus (MERS), influenza type A (InfA), respiratory syncytial virus (RSV), rhinoviruses, adenoviruses and coxsackieviruses. The vast majority of the data summarized in this table come from clinical data from human patients or experiments performed on macrophages, monocytes or dendritic cells isolated from human patients, although some animal-derived data were also consulted where human data were lacking [110,111,112,113,114,115,116,117,118,119,120,121,122,123,124,125,126]. 

A number of patterns emerge from Table 2. Most respiratory viruses activate TLR3, TLR7 and/or TLR8, TLR9, NLRP3, RIG1 and, sometimes, NOD2. This pattern holds, for example, for the influenza A virus [110,111,112,113,114], which notably also downregulated TLR2 and TLR4 expression [111]. Respiratory syncytial virus (RSV) activates TLRs 3 and 7 in both mice and humans [114,115,116,117], and there are some reports of TLR4 activation by RSV in both species [114,118], but these have been contradicted [119] and demonstrated to be due to environmental exposure to bacterial LPS [120]. There is no evidence in humans for the activation or increased expression of TLRs 1, 2 or 6 in RSV infection, and the evidence for the activation of TLRs 8 and 9 in humans is weak [117,118]. RIG1, NOD2 and NLRP3 are, however, clearly activated [121,122]. Adenoviruses [114,123] and coxsackieviruses [124,125,126] have similar receptor activation patterns. 

Rhinoviruses and coronaviruses, however, exhibit some notable differences from the general innate receptor activation pattern for the respiratory viruses just described. Rhinoviruses upregulate TLR3 [127,128,129] and TLR7 [129] but not TLR8 [129] or any other TLR. There is currently no evidence of activation of NLRP3, NOD1 or NOD2 and conflicting reports as to whether RIG1 is activated [128,129]. In consequence, cytokine releases in rhinovirus infections are very limited, especially compared with other respiratory viruses. 

Coronaviruses are also notably different from many other respiratory viruses. While SARS-CoV-2 activates TLR3, TLR7 and NLRP3 like the other viruses [78,130,131,132,133]—as do all coronaviruses [134,135,136,137,138,139,140,141]—coronaviruses express a protein (papain-like protease 1) that strongly antagonizes RIG1 [135,142]. TLR9 is antagonized in SARS-CoV-1 infections and not activated in CoV-229 infections, so it is also unlikely to be activated by SARS-CoV-2 [135]. TLR2, TLR4 and TLR5 are not activated by most coronaviruses [135] and TLR4 is downregulated in MERS [143], but conflicting data exist for whether TLR2 and TLR4 are activated by SARS-CoV-1 in some types of monocytes [144,145]. No coronavirus is known to activate NOD1, but some do activate NOD2 [139,146]; however, data are limited, and more research is needed in this matter, and there appear to be no relevant studies of SARS-CoV-2 yet. Thus, unlike most respiratory viruses, coronaviruses antagonize RIG1 and may not activate, or may even antagonize, TLR9. 

In short, respiratory viruses share a very limited innate receptor activation pattern (IRAP) focused on TLR3, TLR7 and NLRP3 but such viruses can vary considerably in the activation or suppression of TLR2, TLR4, TLR9, RIG1 and NOD2 (Table 2). Notably, none of these activation patterns are identical to, nor are they as diverse as, those summarized in Table 1 for severe COVID-19 or influenza-associated ALI/ARDS patients.

As in the previous section, the data summarized in Table 2 can be integrated into the template provided by Figure 2A,B to provide insight into the sets of synergies and antagonisms that result. As can be seen in Figure 4, coronaviruses, including SARS-CoV-2, activate only a couple of synergistic receptor networks and no (as of the current literature) antagonistic networks. In contrast, Figure 5 illustrates the fact that other respiratory viruses, with the notable exception of rhinoviruses, are likely to activate up to four synergistic receptor networks but, also, one or two antagonistic ones as well. Rhinoviruses activate the least number of innate receptors, resulting in a single synergistic interaction between TLR3 and TLR7. Since rhinoviruses have never been associated with cytokine storms, and most uncomplicated coronavirus and influenza virus infections do not result in cytokine storms, it seems likely that cytokine overproduction requires some minimum number of synergistic PAMP activations that are in excess of three or four. The upshot of these figures is to suggest that there is nothing in the regulatory networks of innate immune system receptors that would lead one to expect the overproduction of cytokines due to an uncomplicated respiratory virus infection, especially one due to a coronaviruses or other uncomplicated viral infections. 

Most importantly, in terms of testing the hypothesis laid out in this paper, Figure 5 (coronavirus IRAP) does not correspond to the synergies and antagonisms present in the severe COVID-19 activation pattern (Figure 3A), nor does Figure 6 (respiratory virus IRAP) correspond to the activation pattern for influenza-associated ALI-ARDS in Figure 3B. Both Figure 5 and Figure 6 differ significantly from that for sepsis patients, as well (Figure 3C). Severe COVID-19, ALI/ARDS and sepsis patients are all characterized by the activation and increased expression of TLR2, TLR4 and NOD2 (and sometimes NOD1, as well) and their associated synergisms (which are quite numerous; see Figure 2A,B and Figure 3A–C). These TLR activations and their associated synergisms are absent from uncomplicated respiratory viral infections of all kinds (Figure 4, Figure 5 and Figure 6). 

Respiratory viruses, in short, do not present the range of PAMPs to the innate immune system necessary to activate the range of receptors characterizing cytokine release syndromes, nor do they result in the sets of synergistic receptor interactions that characterize the TLR–NLR synergy profiles of cytokine release syndromes. It is therefore unlikely that virus infections on their own are responsible for the dysregulation of innate immunity leading to cytokine overproduction syndromes.

### 2.8. Innate Receptor Activation by Bacterial and Fungal Infections Associated with Coronavirus, Influenza and Other ALI/ARDS Syndromes

As noted in Section 2.1 above, severe COVID-19 is highly associated with bacterial and fungal infections, which is also true of influenza-associated ALI-ARDS and polymicrobial sepsis [22,23,24,40,41,42,43,44,45]. The most common bacteria associated with COVID-19 are *Streptococci, Klebsiella pneumoniae, Haemophilus influenzae* and *Mycoplasma pneumoniae*, which are also very common among influenza-associated ALI/ARDS patients [40,41,42,43,44,45]. Fungal infections also occur frequently in severe COVID-19, the most common being the *Aspergillus, Candida* and *Cryptococcus* species [40,41,42,43,44,45]. These clinical findings raise the question of whether bacterial or fungal infections might, in and of themselves, be responsible for the cytokine storms found in COVID-19, influenza-associated ALI-ARDS and sepsis.

Table 3 summarizes studies of innate immune system receptor activation caused by the bacterial pathogens most commonly associated with COVID-19 and influenza-ALI/ARDS. Table 4 summarizes similar data for the most common fungal infections associated with these syndromes. Bacteria generally activate TLR1, TLR2, TLR4, TLR9, NLRP3 and NOD2 (Table 3) (reviewed in [147], Group A Streptococcal activation [148,149,150,151,152,153,154,155,156], Group B Streptococcal activation [68,147,157,158,159], Staphylococci [147,151,160,161], Mycobacteria [147,162,163], Klebsiella [164,165,166,167,168,169], Haemophilus [147,149,150,170,171,172,173,174], Legionella [147,163,175,176], Chlamydia [147,163,177,178,179,180], Neisseria [147,149,181,182], Pseudomonas [147,163,183,184,185] and Mycoplasma [98,147,186,187,188]). Gram-negative bacteria activate NOD1 as well, because they express meso-DAP as part of their cell walls, which Gram-positive bacteria do not, because they lack this molecular constituent. Mycobacteria tend to be ambiguous on Gram testing due to unusual cell wall structures [147], and Mycoplasmas have no cell walls and, therefore, express no PAMPs capable of activating either NOD1 or NOD2. Not surprisingly, the main virus-activated receptors TLR3, TLR7, TLR8 and RIG1 are not activated by bacteria, but, notably, most respiratory bacteria do activate TLR9 by means of the release of mitochondrial DNA [147]. Some individual species of bacteria vary the common theme by also activating TLR5, TLR6, TLR7 or TLR8 (e.g., [157,183] (Table 3), but these TLR are rarely studied in the bacterial activation of the innate immune system, so it is not known how common such activation may be.

The activation of innate immune receptors by fungi (reviewed in [189,190,191,192]) generally follows the same pattern as bacteria, with TLR2, TLR4, TLR9 and NLRP3 being common to the fungi (*Aspergillus, Candida* and *Cryptococcus* species) most often associated with COVID-19, ALI/ARDS and sepsis. Again, like bacteria, individual species of fungi can also activate a range of other TLR and NOD.

As before, integrating the data from Table 3 and Table 4 into Figure 2A,B yields a consensus diagram (Figure 7) of the activation patterns of bacteria and some fungi (particularly Aspergillus species). Figure 7 demonstrates that bacterial and fungal PAMPs are likely to activate up to six synergistic receptor interactions balanced by up to six antagonisms that may moderate receptor activity. This activation pattern would explain how systemic bacterial infections induce significant cytokine release, leading to fever, chills and joint soreness, among other symptoms.

It must again be asked whether the pathways illustrated in Figure 7 correspond to those characterizing severe COVID-19, influenza-associated ALI/ARDS or sepsis. The reference to Figure 3A–C demonstrates that the set of TLR and NLR activated by bacteria do not match those activated in severe COVID-19, influenza-associated ALI-ARD or sepsis (compare also Table 3 and Table 4 (bacterial and fungal activations) with Table 1 (cytokine release syndrome activations)). In particular, severe COVID-19 patients (as well as influenza-associated ALI/ARDS) are characterized by the activation and increased expression of TLR3 and TLR7—two virus-activated receptors—while NOD2 does not appear to play a major role. Thus, as with viral infections, simple mono-infections with bacteria or fungi do not appear to be able to account for the IRAP that characterize cytokine release syndromes.

### 2.9. Do Combinations of Viruses and Bacteria Explain Innate Receptor Activation Patterns in Severe COVID-19 and ALI/ARDS?

If neither viral activation patterns of innate immunity (Section 2.7) nor bacterial/fungal activation patterns of innate immunity (Section 2.7) display the range of activated TLR and NLR to reflect the innate immunity activation patterns of either severe COVID-19 or influenza-associated ALI/ARDS, do combinations of viruses with bacteria or fungi do so, as postulated in the hypothesis above (Hypothesis, Section 2.4)?

Table 5 presents a summary of the consensus activation patterns of microbes derived from the previous tables (Table 1, Table 2, Table 3 and Table 4). Table 5 emphasizes the point that the activation patterns found in severe COVID-19 and in influenza-associated ALI/ARDS display characteristics of both viruses and bacteria and/or fungi. TLR3 and TLR7 activation in severe COVID-19 and influenza-associated ALI/ARDS requires viral activation, while TLR4 and NOD1 or NOD2 indicate bacterial or fungal activation. Thus, it is very likely that both severe COVID-19 and influenza-related ALI/ARDS are the results of multiple, concurrent infections. Adding the virus IRAP in Table 5 to the bacterial and/or fungal IRAP yields activation patterns in general agreement with severe COVID-19 and influenza-associated ALI/ARDS. However, Table 5 also illustrates the fact that these activation patterns are not the simple result of adding bacterial or fungal IRAP to viral IRAP: TLR9 and NOD2 activation by viruses and bacteria is questionable or absent in severe COVID-19 and influenza associated ALI/ARDS, as is the activation of TLR2 and RIG1 that would be expected to be upregulated in influenza-associated ALI/ARDS. 

Consider Figure 8 as an example. Figure 8 describes the synergisms and antagonisms expected from a combination of SARS-CoV-2 with the consensus IRAP for bacteria (which includes Aspergillus fungi species). The results reasonably accurately reflect the actual findings reported for severe COVID-19 patients (Table 1 and Figure 3A) in which TLR2, TLR3, TLR4, TLR7, TLR9 and NLRP3 are activated. Detailed differences would be expected to exist among COVID-19 patients depending on what specific bacteria (Gram-negative, Gram-positive, *Mycobacterial* or *Mycoplasmal*) (Table 3) or fungi (*Candida, Cryptococcus* or *Aspergillus*) (Table 4) were present as coinfections. The result is eight sets of TLR/NLR synergisms but, also, six sets of antagonisms. The two antagonisms of TLR7 and NOD2 acting on TLR9 may, for example, block its activation, resulting in the questionable role of TLR9 in severe COVID-19 (Table 1 and Table 4) through the elimination of two synergisms. However, most respiratory bacteria (with the exception of Mycoplasmas) and fungi activate NOD2, and Gram-negative bacteria activate NOD1, so one would expect to see one or both of the NOD activated in severe COVID-19, which does not appear to be the case (Table 1 and Table 5). It must be presumed that there exist a set of antagonisms acting upon NOD1 and NOD2 that negate their activation in the presence of viral infections, probably through TLR3, TLR7 and or TLR9. The antagonistic actions of NOD2 on TLR2 and TLR4 may also work in reverse given sufficient time for the system of interactions to equilibrate. For example, bacterial outer membrane protein vesicles protect against H1N1, H5N2 and H5N1 and MERS fatal infections in mice [193,194,195] and antagonizing TLR4 blocks the cytokine storm associated with influenza virus infection and improves survival in mice [196]. Given the paucity of research that has so far been published on the activation (or lack thereof) of NLR in severe COVID-19, it is also possible that additional studies will find that NOD1 and/or NOD2 are actually activated or their expression increased during the development of cytokine storms. Alternatively, various negative feedback loops missing in Figure 2A,B may exist that are essential for understanding IRAP in cytokine storm syndromes.

A further reason for thinking that there are likely to be as-yet-unidentified negative feedback systems antagonizing NOD2 and that the antagonistic effects of NOD2 on TLR4 are important in severe COVID-19 is that the interferon (IFN) function is very severely impaired [15,16,197,198,199,200]. Although some of this impairment is certainly mediated by the downregulation of RIG1 by coronaviruses [135,142], NOD2, TLR3 and TLR4 can join RIG-1 in stimulating IFN release (Figure 1). Severe IFN impairment would seem to call for the impairment of more than just one of these pathways. Such a mechanism would, not incidentally, address an ongoing criticism of the role of cytokine storms in severe COVID-19, which is that severe COVID-19 is not actually a cytokine release syndrome, since IFN is severely impaired, but, actually, an immunosuppressive disease in which the virus is enabled to spread to, and replicate in, an uncontrolled fashion in multiple organ systems [198]. In fact, severe COVID-19 may be considered to be characterized simultaneously as immunosuppressive with regards to viral immunity and as a cytokine storm with regards to MyD88-associated cytokine release. Thus, both the immunosuppression (of MyD88-associated pathways) and immunostimulation (of IFN pathways) might be needed [200]. 

One other caveat also needs to be considered in evaluating and comparing Figure 8 to Figure 3A, which is that Figure 3A is derived from studies of the status of TLR and NLR activation in the midst of established severe COVID-19 (i.e., from hospitalized patients), while Figure 8 represents the status of TLR and NLR activation at the onset of infection. Since antagonistic TLR and NLR interactions tend to be initiated over time measured by many hours or several days (see Section 2.3), Figure 8 may well explain how the cytokine storm in severe COVID-19 is initiated and Figure 3A the status of the innate system as it attempts to moderate its cytokine production. One final complication is that many severe COVID-19 patients are treated with multiple antibiotics [201,202,203,204,205,206,207,208,209,210,211,212], which may further modify the expression of bacterially activated TLR and NOD1/NOD2.

The type of variation just described within severe COVID-19 patients coinfected with SARS-CoV-2 and various bacteria and/or fungi applies to an even greater extent to understanding the etiologies of influenza-associated ALI/ARDS and sepsis. Investigators have been well-aware since the 1930s that influenza A virus infections can set the stage for a wide range of bacterial infections with both Gram-positive and Gram-negative bacteria, as well as mycoplasmas such as *Mycoplasma pneumoniae* that can lead to ALI/ARDS [22,23,213,214,215,216]. Figure 9 illustrates the innate receptor activation pattern (IRAP) that might be expected to result from a *Haemophilus influenzae* superinfection of the influenza A virus [22,213,214]. The result is an even more complex set of synergisms and antagonisms than that illustrated above for severe COVID-19. This increased complexity of synergies may help to explain why ALI/ARDS is typically characterized by higher levels of cytokine releases than in severe COVID-19 (see Section 2.3). As with the severe COVID-19 case just described, however, it must again be pointed out that Figure 9 is more complex than the IRAP that is actually observed in influenza-associated ALI/ARDS (Figure 3B), probably for the same reasons just discussed for severe COVID-19. Figure 9 represents the initiation of TLR/NLR activation, while Figure 3B represents the established pattern after antagonistic feedback; antibiotic treatments undoubtedly modulate bacterial PAMP presentation to TLR and NOD, and there are likely to be as-yet-uncharacterized negative feedback effects of TLR2, TLR4, etc. on NOD1/NOD2 [193,194,195,196] and, especially, in this case, RIG1, which would otherwise be predicted to be upregulated by the influenza virus (Table 2).

Finally, Figure 10 illustrates the sets of TLR and NLR synergisms and antagonisms that might be expected from a combined infection with a Gram-positive and a Gram-negative bacterium, as might occur during polymicrobial sepsis. Once again, the number of synergisms and antagonisms is large, the exact homeostatic balance achieved difficult to predict and, in its details, dependent on the TLR activated by the specific bacteria (and/or fungi) involved, which might include TLR1, TLR5 and TLR6 (Table 3). In comparing Figure 10 to Figure 3C (known TLR and NLR activation patterns in human sepsis patients), all of the caveats just stated for severe COVID-19 and ALI/ARDS patients once again apply; Figure 10 describes the initial effects of combined bacterial and/or fungal infections on TLR and NLR, whereas Figure 3C represents the immune status during hospitalization, and there are likely negative feedback effects that are missing, etc. Nonetheless, Figure 10 certainly suggests why combined infections would be more likely to result in a cytokine storm during sepsis than would a mono-infection with a single bacterial or fungal species (Figure 7).

The most important takeaway from the current section is that the only way to explain the diversity of TLR and NLR activations found in Figure 3A–C is to consider them to be the result of varied combinations of infections, as proposed in the hypothesis (Section 2.4 above). Since SARS-CoV-2 and influenza viruses activate different sets of TLR and NLR and may synergize with different bacteria, the cytokine release syndromes that result differ. Sepsis differs as well, probably because viruses do not play a major role in its initiation; sepsis seems to be explained best by combinations of bacteria and, possibly, fungi. Synergisms are not, however, the whole story. In order to account for the specific details of innate immune activation profiles described in Figure 3A–C, it is very likely that negative or antagonistic feedback systems must come into play as the disease progresses. Some of these feedback systems must involve the negative regulation of NOD1, NOD2 and RIG1, since these receptors are known to be activated by the viral and bacterial infections discussed here, yet they do not appear to be activated during cytokine release syndromes. At present, this makes no sense from any existing perspective, since RIG1 should be either upregulated or downregulated by any viral infection (Table 2), and NOD1 and NOD2 should be stimulated by the vast majority of bacterial infections (Table 3). Since there is, at present, almost no evidence of the negative regulation of these receptors (Figure 2A,B), the apparent lack of any known role for NOD1, NOD2 and RIG1 in established cytokine release syndromes therefore represents an important anomaly in need of exploration.

### 2.10. Role of DAMPs in Driving COVID-19 and Other Cytokine Release Syndromes

One additional explanation for the lack of NOD1, NOD2 and RIG1 activation in cytokine release syndromes may come from consideration of the role that DAMPs play maintaining a cytokine release following its initiation by viral, bacterial and fungal stimulation. DAMPs, as noted in Section 2.1, are molecules released by host cells as danger signals following cellular damage from infection or injury. While Figure 8, Figure 9 and Figure 10 represent PAMP-induced IRAP, Figure 3A–C may represent DAMP-maintained IRAP that remain after the immune system, often supported by appropriate hospital treatments, has controlled or moderated the initial infections and their expression of PAMPs. 

DAMPs probably play a significant role in supporting cytokine storms, but comparatively little research has been expended in studying DAMPs as compared to PAMPs in this context, so the details of what roles DAMPs are playing in maintaining cytokine overproduction syndromes and how to intervene in their effects are vague [217,218]. Critically ill COVID-19 patients, ALI/ARDS patients and sepsis patients all release high levels of extracellular histones (TLR2, TLR4 and NLRP3 activators [219]); neutrophil elastase (a TLR4 activator [220]) and cell-free DNA (a TLR9 activator [221]), each of which correlate with the probability of the patients being admitted to the intensive care unit and of dying [222]. The nuclear protein high mobility group box 1 protein (HMGB1) is also released during severe COVID-19, ALI/ARDS and sepsis and drives TLR2 and TLR4 activation [223]. Additionally, the heat shock protein HSP5a (also known as GRP78), a marker for endoplasmic reticulum stress and the TLR2/TLR4 activator, is released at unusually high concentrations in severe COVID-19 patients [224,225]. Other possible DAMPs that may be playing roles in COVID-19 include extracellular RNAs [226] that can activate TLR3 and TLR7, as well as extracellular hemoglobin, which activates NLRP3 and is associated with coagulation dysregulation and microclotting [227,228]. In sum, the activation of TLR2, TLR4, TLR7, TLR9 and NLRP3 are almost certainly maintained by DAMPs in severe COVID-19, ALI/ARDS and sepsis and may need to be therapeutically addressed [217,218,226]. Notably, DAMPs are not known to activate NOD1, NOD2 or RIG1, which might help to explain the absence of their observed activation in established cytokine storm syndromes (Figure 3A–C) as compared with their initial stimulating conditions (Figure 8, Figure 9 and Figure 10).

## 3. Discussion 

### 3.1. Summary of the Synergistic Activation of TLR and NLR in Cytokine Storm Syndromes

The object of this study was to provide a model of why uncomplicated SARS-CoV-2 infections do not result in the “cytokine storm” that characterizes severe COVID-19 accompanied by ARDS and to explore what differentiates severe COVID-19-associated cytokine release syndrome from those associated with ALI/ARDS following influenza A infection and from sepsis. In severe COVID-19, it is proposed that cytokine storms result from synergistic interactions among Toll-like receptors (TLR) and nucleotide-binding oligomerization domain-like receptors (NLR) due to combined infections of SARS-CoV-2 with other microbes, mainly bacterial and fungal. This proposition is based on nine linked types of evidence and their logical connections: (1) Severe cases of COVID-19 differ from healthy controls and mild COVID-19 patients in exhibiting increased TLR4, TLR7, TLR9 and NLRP3 activity. (2) SARS-CoV-2 and related coronaviruses activate only TLR3, TLR7, RIG1 and NLRP3. (3) SARS-CoV-2 cannot, therefore, account for the innate receptor activation pattern (IRAP) found in severe COVID-19 patients. (4) Severe COVID-19 also differs from its mild form in being characterized by bacterial and fungal infections. (5) Respiratory bacterial and fungal infections generally activate TLR2, TLR4, TLR9 and NLRP3. (6) A combination of SARS-CoV-2 with bacterial/fungal coinfections accounts for the IRAP found in severe COVID-19 and how it differs from mild cases. (7) Many pairs of TLR and NLR synergize so that combined infections can greatly enhance the cytokine release. For example, in severe COVID-19, TLR7 (viral) and TLR4 (bacterial/fungal) synergize, TLR9 and TLR4 (both bacterial/fungal) synergize and TLR2 and TLR4 (both bacterial/fungal) synergize with NLRP3 (viral and bacterial). (8) Thus, a SARS-CoV-2 bacterium/fungus coinfection produces a synergistic innate activation resulting in the hyperinflammation characteristic of a cytokine storm. However, different respiratory bacteria or fungi will activate different sets of synergistic TLR–TLR, TLR–NLR or NLR–NLR pairs, resulting in different expressions of cytokine overproduction. (9) Since SARS-CoV-2 activates a different set of TLR and NLR than other respiratory viruses such as influenza A virus or the respiratory syncytial virus do, the presence of coinfecting bacteria or fungi will result in different cytokine release profiles than those seen in COVID-19, thereby explaining the variability observed in cytokine release syndromes.

In sum, while there is a natural tendency to ascribe to all symptoms of a new disease to whatever new agent is discovered to be its cause, it is often the case that variations in the expression of an infection are due to additional host factors. Even the founders of the germ theory of disease, Louis Pasteur and Robert Koch, understood that, in Pasteur’s words, “the terrain is as important as the germ”, by which he meant that the health of the host at the time of infection can alter significantly the course of a disease and its symptoms. This “terrain” concept provides an appropriate context for understanding the causes of cytokine storms and the difficulties that exist in defining and characterizing such “storms” unambiguously. The proposition that guided this review is that cytokine storms do not occur in mono-infected individuals but only in those with multiple, concurrent infections, and furthermore, the specific nature of the cytokine release syndrome is determined by the particular set of infections that afflict the host. Thus, uncomplicated SARS-CoV-2 infections cannot lead to cytokine release syndrome (Table 2 and Figure 4) any more than uncomplicated influenza A virus infections can (Table 2 and Figure 5). The sets of TLR–TLR, TLR–NLR and NLR–NLR synergisms are too few to account for such overproductions, particularly when antagonistic interactions negatively controlling cytokine production are taken into account. It is concluded that SARS-CoV-2 and influenza A virus are not sufficient causes of these syndromes but require the presence of additional, concurrent bacterial or fungal infections that enhance their innate activation profiles. 

Moreover, an essential part of the model is the principle that different sets of coinfections will result in different innate activation profiles and, thus, different forms of cytokine overproduction syndromes. Among COVID-19 ARDS, patients can have different etiologies due to superinfections with a variety of Gram-positive or Gram-negative bacterial, mycoplasmal or fungal infections, each resulting in a different cytokine overproduction profile. Moreover, COVID-19 ARDS patients have different etiologies and cytokine overproduction profiles than influenza-associated ALI/ARDS patients do, who differ as well from sepsis patients. In this way, the model presented here explains why there is not, and can never be, a universal definition of a “cytokine storm”, nor a single set of clinical criteria for diagnosis. Each patient represents a unique set of risk factors for both viral infection and for bacterial or fungal superinfections, and the consequent mix of risk factors and infections results in unique innate activation profiles (IRAP). Determining the specific IRAP for each individual patient may be the key to learning how to best treat the range of cytokine overproduction syndromes.

### 3.2. The Role of Pathogen Synergisms in COVID-19

Evidence for the presence of bacterial and fungal coinfections in severe COVID-19 was provided in Section 2.1 above and is consistent with other clinical observations as well, such as reports that the SARS-CoV-2 viral load is not directly correlated with COVID-19 severity. For example, Zhou et al. [229] demonstrated that, in severe COVID-19 patients coinfected with *S. pneumoniae, Haemophilus parainfluenzae* or *Neisseria meningitides*, the numbers of SARS-CoV-2 virions were significantly lower than in uncomplicated SARS-CoV-2 infections. Similarly, Schlesinger et al. [230] reported that, among COVID-19 patients admitted to the intensive care unit, “No difference of viral load was found in tracheal or blood samples with regard to 30-day survival or disease severity. SARS-CoV-2 was never found in dialysate. Serologic testing revealed significantly lower concentrations of SARS-CoV-2 neutralizing IgM and IgA antibodies in survivors compared to non-survivors (*p* = 0.009).” These counterintuitive data again argue against SARS-CoV-2 being the sole driver of the rates of morbidity and mortality among COVID-19 patients and for additional factors that were not measured, such as the presence or absence of neutralizing IgM and IgA against bacterial or fungal coinfections. Animal models also support this “terrain” concept. In murine models of fatal coronavirus (MERS and SARS) and influenza pneumonias, TLR3–TLR4 synergism is universal, such that antagonizing or deleting TLR4, which is specifically activated by bacterial lipopolysaccharides, prevents death [231,232,233,234]. Indeed, there are multiple reports that vaccinations against *S. pneumonia* [235,236,237,238,239,240,241,242] and, possibly, *Haemophilus influenzae* as well [239,240,241] significantly decreases the probability of contracting and dying from COVID-19. In this context, it is too bad that there are not vaccines against *Klebsiella pneumoniae, Mycoplasma pneumoniae* and *Aspergillus* fungal infections, as well.

However, it must be noted that, against the bacterial and fungal coinfection interpretation of disease severity just presented, Blot et al. [243] and Bitker et al. [244] found that SARS-CoV-2 viral loads did predict adverse outcomes among severe COVID-19 patients, perhaps suggesting that, in the absence of synergistic coinfections, other host risk factors such as obesity, diabetes, heart disease and smoking can also affect the innate immune system function in such a way as to provide “fertile terrain” for the virus. For example, it is known that obese individuals have increased susceptibility to serious or fatal influenza [245]. Clearly, more studies of the virion and antibody prevalence performed in the context of the “terrain” concept are needed to sort out this question.

The IRAP model also provides a basis for understanding why some viruses are associated with cytokine storms or release syndromes, such as coronaviruses, influenza viruses and respiratory syncytial virus, while others, such as rhinoviruses, are not. References to Table 2 and Figure 4 illustrate the fact that the range of TLR and NLR activated by rhinoviruses is so limited as to make it difficult for these viruses to participate in interactions with other pathogens that could result in the necessary synergistic pathway activation necessary for the overproduction of cytokines. Thus, not all virus-bacteria coinfections will necessarily result in cytokine overproduction syndromes.

### 3.3. Implications of Innate Receptor Activation Profiles for Tretment of COVID-19 and Other Cytokine Release Syndromes

Perhaps the most important aspect of the IRAP model produced is to provide clues as to how to treat the variety of cytokine release syndromes more effectively than is currently possible. To begin with, the model suggests that there is no single set of TLR and NLR that encompasses the range of TLR and NLR activated in such syndromes. The only set shared by severe COVID-19, influenza-associated ALI/ARDS and sepsis is that comprised of TLR4, TLR7 and NLRP3. It would therefore seem that these three receptors would, at a minimum, need to be addressed concurrently to downregulated cytokine release in any of these syndromes. There appears to be no standard approach to treating cytokine release syndromes that currently addresses all three, and the vast majority address none, focusing instead on antagonizing cytokine functions. However, melatonin (N-acetyl-5-methoxy tryptamine) has recently emerged as an effective therapy for severe COVID-19 [246,247,248,249,250,251]. While melatonin is often thought of being mainly a sleep regulator, it also has widespread effects on the innate immunity—in particular, antagonizing TLR2, TLR4, TLR9, NLRP3 and, possibly, NOD2 [59,252,253,254,255]. Melatonin therefore antagonizes five of the seven TLR and NLR activated in COVID-19, eliminating all but one of the receptor synergisms (Figure 11). Additionally, all of the pathways resulting in interferon, interleukin and TNF release are moderated, causing a global decrease in cytokines (Figure 12). The effectiveness of melatonin for treating COVID-19 cytokine storms is therefore easily explained by the synergistic activation IRAP model of severe COVID-19 proposed here.

The model of innate immunity regulation proposed here may also be useful for evaluating the likelihood that other therapies will be useful in treating severe COVID-19 and, particularly, the cytokine storm accompanying it. Consider, as examples, Anakinra, a human IL-1 receptor antagonist, and Tocilizumab and Sarilumab, monoclonal antibodies that block the IL-6 receptor and, thus, its signal transduction pathway [256,257]. Each one neutralizes only one type of cytokine among the dozen that are released during a cytokine storm (Figure 12). Thus, one would expect each one to have a minimal effect on COVID-19 symptoms and progression, which is, in fact, the case. While a meta-study of Tocilizumab found a weak but statistically significant reduction in mortality among ventilated patients [258], the largest double-blinded, randomized study for treating COVID-19 patients found no benefits compared to the standard care [259]. A head-to-head comparison of Anakinra to Tocilizumab found no difference in patient outcomes [260], which in light of the lack of Tocilizumab benefits, indicates a lack of Anakinra benefits, as well. Anakinra plus Tocilizumab, however, reduced the need for ventilation and overall mortality [261], as might be expected, since the combination addressed two sets of cytokines rather than just one (Figure 12). Targeting a broader range of cytokines receptors might be more therapeutically effective but, also, much more difficult to deliver than targeting key TLR and NLR and, at present, extremely expensive. 

Results such as these suggest, in light of the IRAP model presented here, that the most effective treatments for cytokine storms, whether in COVID-19 or other cytokine release syndromes, are most likely to be those that target either the key TLR–NLR synergistic sets that trigger a particular syndrome through broadly acting inhibitors such as melatonin or, perhaps, colchicine (which has a similar TLR antagonism profile [262,263,264,265,266]) or by means of TLR and/or antagonist combinations, thereby downregulating all cytokine production. It does not appear that either of these approaches has been systematically investigated in COVID-19, ALI-ARDS or sepsis, and even the use of melatonin in COVID-19 was initiated in apparent ignorance of its multiple TLR/NLR-moderating effects. Since many individual TLR and NLR antagonists are being investigated, and some are in clinical trials, (reviewed in [267]), such multipronged therapies are quite feasible. 

It is important to emphasize again that effective therapies need to be tailored to the specific etiologies of each patient’s syndrome, and therefore, it is essential that every ALI/ARDS or sepsis patient be tested for the specific set of coinfections that might be present so that the particular TLR/NLR activation profile can be predicted. The literature references in Section 2.1 makes it clear that such a diagnosis should not involve attempts to isolate such coinfections from the blood, sputum or biopsy, which often fail to identify such infections, but should utilize highly sensitive urine antigen tests and/or polymerase chain reaction tests.

One final implication of the present review is that it is not appropriate to attempt to treat COVID-19-associated cytokine release in the same way as influenza-associated ALI/ARDS or sepsis-related cytokine storms. As noted in several places above (most notably in the Introduction and Section 2.8), interferon release is severely impaired in COVID-19, which facilitates SARS-CoV-2 infectivity and replication. A number of groups have therefore suggested, correctly in light of the IRAP model, that patients with severe COVID-19 need interferon supplementation [267,268,269,270,271,272], a therapeutic approach that would very likely be harmful in other cytokine release syndromes in which interferon is overproduced. An alternative suggested by the IRAP model might be to stimulate a natural interferon release using RIG1, NOD1 and, perhaps, TLR3 agonists (Figure 12). Indeed, a just-released clinical study demonstrated that COVID-19 patients treated with the hepatitis drug peg-interferon are four times more likely to recover quickly and clear their SARS-CoV-2 infection than equivalently treated patients not given that drug [273]. 

### 3.4. Directions for Future Research

There is a great deal that is not known about the innate receptor activation patterns (IRAP) of the pathogens associated with cytokine overproduction syndromes, as a glance at any of the blank entries of the tables in this paper makes clear. Each of these blank spots represents an instance of ignorance that needs to be rectified by future research. While these blank spots are too numerous to discuss in detail, certain generalizations are possible.

One of the most pressing needs is the further investigation of antagonisms between TLR and other TLR, TLR and NLR and between NLR and other NLR. For whatever reason, synergisms have been the focus of many more studies than antagonisms (perhaps because the latter require chronic stimulation to become apparent), yet one of the key findings of this paper is that it is difficult to make sense of the IRAP characterizing severe COVID-19, influenza-associated ALI/ARDS or sepsis in terms of their probable multifactorial etiologies simply in terms of their known receptor antagonisms. It seems very likely that NLRP3, NOD1 and RIG1 antagonize several TLR and possibly each other as well, just as is known for NOD2 (Figure 2B). 

Another pressing need is for more complete controls in TLR and NLR studies to definitively rule out the activation of receptors that are currently assumed to be irrelevant. An all-too-common approach to TLR and NLR research that became apparent in reviewing this literature was a habit to choose one or a few TLR or NLR to study in any given system without employing any controls, negative or positive. The absence of data on the activation or antagonism of TLRS 1, 5 and 6 across almost all of the types of studies employed in this paper, as evidenced by the very numerous blanks in almost all of the tables, is particularly noteworthy.

Yet another major gap in our current understanding of cytokine overproduction syndromes is the role that DAMPs may play, possibly in initiating, and almost certainly in maintaining cytokine release. Compared with the number of studies on PAMP activation of TLR and NLR, the literature on DAMP activation is extremely sparse and there is virtually no literature on whether DAMP activation of TLR and NLR can result in antagonisms between these receptors. Such studies may provide clues concerning novel types of DAMP-derived therapies for limiting or controlling cytokine overproduction. 

New types of animal models are suggested by the IRAP model presented here. Among the possibilities are animal models of COVID-19 employing combinations of SARS-CoV-2 with *Streptococci, Klebsiella, Mycoplasmas, Aspergillus* or other commonly occurring coinfections associated with the severe form of the disease. These combinations should, if the IRAP model presented here is correct, induce a super-additive cytokine release compared with SARS-CoV-2 alone or the coinfection alone. Additionally, such models could shed light on the mechanisms of severe complications, such as blood coagulation disorders, cardiovascular pathologies and respiratory distress that accompany severe COVID-19.

Finally, the IRAP model suggests a very different approach to developing drugs to treat cytokine overproduction syndromes than is generally used. In the first place, as noted above in Section 3.3, not all cytokine overproduction syndromes have the same etiologies or mechanisms of cytokine release, so a one-size-fits-all approach is unlikely to be successful. Secondly, most of the current approaches to cytokine control seem to be targeted at antagonizing individual cytokines, such as IL-6 or the TNFs, and mainly by means of blocking their receptors, while the IRAP model suggests that a much more effective approach would be to search for broadly acting TLR or NLR antagonists that will prevent the release of all or most cytokines. The possibility that melatonin and colchicine may act in this way provides hope that such an approach is not only possible but likely to be achievable. 

### 3.5. Limitations and Sources of Bias in This Study

In conclusion, it is important to point out the possible limitations and sources of bias in this study. As noted above in Section 2.5, no attempt was made to perform a complete literature search on all aspects of cytokine release syndrome-related studies related to innate receptor activation. The studies reviewed and the data from which were incorporated into the model presented here were chosen for their relevance to testing the alternative hypotheses laid out in Section 2.4 and differentiating between their predictions. Thus, the data incorporated here is limited to studies relevant to TLR–TLR, TLR–NLR and NLR–NLR synergisms and antagonisms; the activation profiles of the most common pathogens associated with COVID-19, influenza-associated ALI/ARDS and sepsis; the actual activation patterns observed in human patients with those cytokine overproduction syndromes and what is known about the effects on TLR and NLR activation caused by a very select group of therapies used for treating COVID-19 and other cytokine overproduction syndromes. It is quite possible that this focus on testing the particular hypotheses described in Section 2.4 has resulted in overlooking important and relevant studies that could support other explanations for cytokine overproduction and result in different models and predictions about therapies. Other innate receptors not included in this model and their associated pathways undoubtedly also play important roles in cytokine overproduction that may nuance or modify the IRAP model. Such myopia is a risk associated with any type of model-building or hypothesizing. Finally, it is important to stress the incompleteness of the data used in this study, as is all-too apparent in the many blank spaces in the tables. There is a great deal that we still do not know about TLR and NLR activation patterns, synergisms and antagonisms that could modify or even overturn the IRAP model presented here. 

## 4. Conclusions

Evidence was presented that sepsis, ALI, ARDS and severe COVID-19 are all characterized by being polymicrobial infections, which is likely to be true of all instances of cytokine storms or cytokine release syndromes. These polymicrobial infections express PAMPs that synergistically activate various sets of TLR, NLR and RIG that produce the particular expression of cytokines characterizing the overproduction syndromes. The particular sets of TLR, NLR and RIG that are activated by the particular combination of infectious agents result in the release of different distributions of cytokines in each instance—most notably, a significant impairment of interferons in COVID-19 that is not typical of other cytokine storm syndromes. In other words, there is no single entity that can be defined as a “cytokine storm” or “cytokine release syndrome”, because different combinations of pathogens will induce different distributions of cytokines. Consequently, there can also be no single therapeutic approach to the hyperinflammation that follows from the overproduction of cytokines; rather, therapies must be tailored to the sets of receptors activated in any given disease, which requires identifying the etiological factors at work in each instance and/or the particular IRAP of each individual patient. That said, however, it will be generally true that effective therapies will be those that antagonize as many of the receptors activated synergistically in each syndrome as is possible, so that broadly acting TLR and NLR antagonists are likely to be more effective than specifically targeted ones or therapies targeted at specific cytokines or their receptors. A better understanding of the innate receptor activation patterns characterizing various cytokine release syndromes, the specific PAMPs and DAMPs driving their activation and the methods for quickly screening patients for TLR and NLR activation may result in better targeted, individualized treatments. 

## Figures and Tables

**Figure 1 ijms-22-02108-f001:**
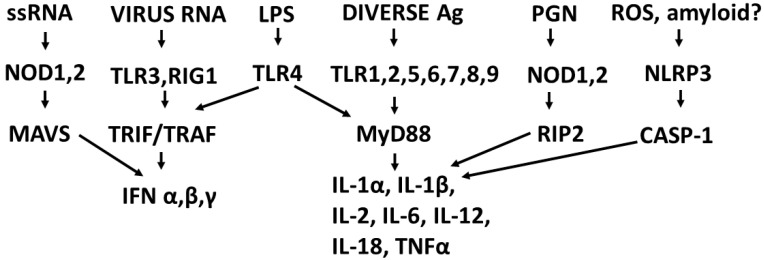
Outline of some of the main signaling pathways leading to cytokine release. RNA = ribonucleic acid; ss = single-stranded; LPS = lipopolysaccharide; Ag = antigens; PGN = peptidoglycans; ROS = reactive oxygen species; NOD = nucleotide-binding oligomerization domain-containing protein; TLR = Toll-like receptor; RIG1 = retinoic acid inducible gene protein-1; NLRP3 = NOD-, LRR- and pyrin domain-containing protein 3; MAVS = mitochondrial antiviral-signaling protein; TRIF = TIR domain-containing adapter-inducing interferon-β; TRAF = TNF receptor-associated factor; MyD88 = myeloid differentiation primary response 88 protein; RIP2 = receptor-interacting-serine/threonine-protein kinase 2; CASP-1 = Caspase-1 or the interleukin-1 converting enzyme; IFN = interferon; IL = interleukin and TNF = tumor necrosis factor.

**Figure 2 ijms-22-02108-f002:**
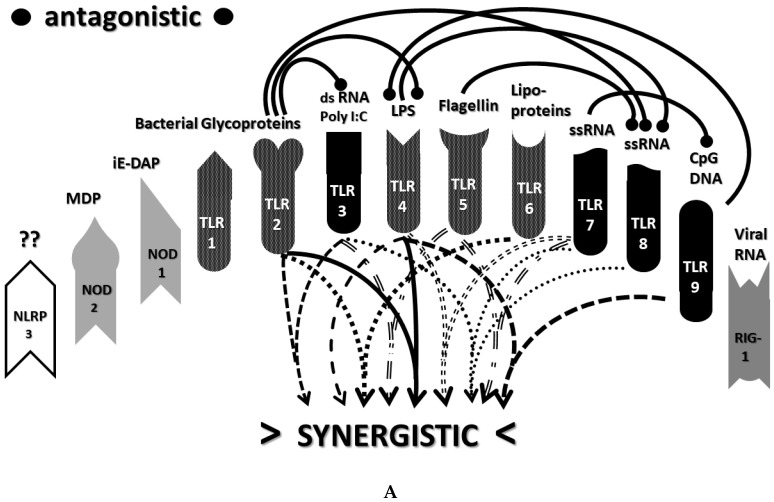
(**A**) Known synergistic and antagonistic interactions among TLR. Antagonistic effects are illustrated so that the receptor that is antagonized has a large dot above it. The result is a very complex system of feedforward and feedback loops, many of which involve bacterial antigens (TLR1,2,4, 5 and 6) with each other and with viral antigens (TLR3, 7, 8 and 9). Receptors activated by viral antigens also tend to synergize with each other. This figure is based on less-complete versions in [51,56] that were augmented with reference to the following studies [57,58,59,60,61,62,63,64,65,66,67,68,69,70,71,72,73,74,75]. The different line styles have no intrinsic meaning but are used to help the reader more easily trace the pairs of interactions, in the case of synergisms, and the source and direction of the interactions, in the case of antagonisms. (**B**) Known synergistic and antagonistic interactions between NLR, NLRP3 and RIG1 and with TLR. Antagonistic effects are illustrated so that the receptor that is antagonized has a large dot above it. The result is a very complex system of feedforward and feedback loops, many of which involve bacterial antigens with each other (TLR2 and 4 and NOD 1 and 2), bacterial antigens with viral antigens (NOD1 and NOD2 with TLR9 and TLR3 with RIG1) and between viral antigens (TLR3 and RIG1). This figure is based on less-complete versions in [51,56] that were augmented with reference to the following studies [57,58,59,60,61,62,63,64,65,66,67,68,69,70,71,72,73,74,75]. The different line styles have no intrinsic meaning but are used to help the reader more easily trace the pairs of interactions, in the case of synergisms, and the source and direction of the interactions, in the case of antagonisms.

**Figure 3 ijms-22-02108-f003:**
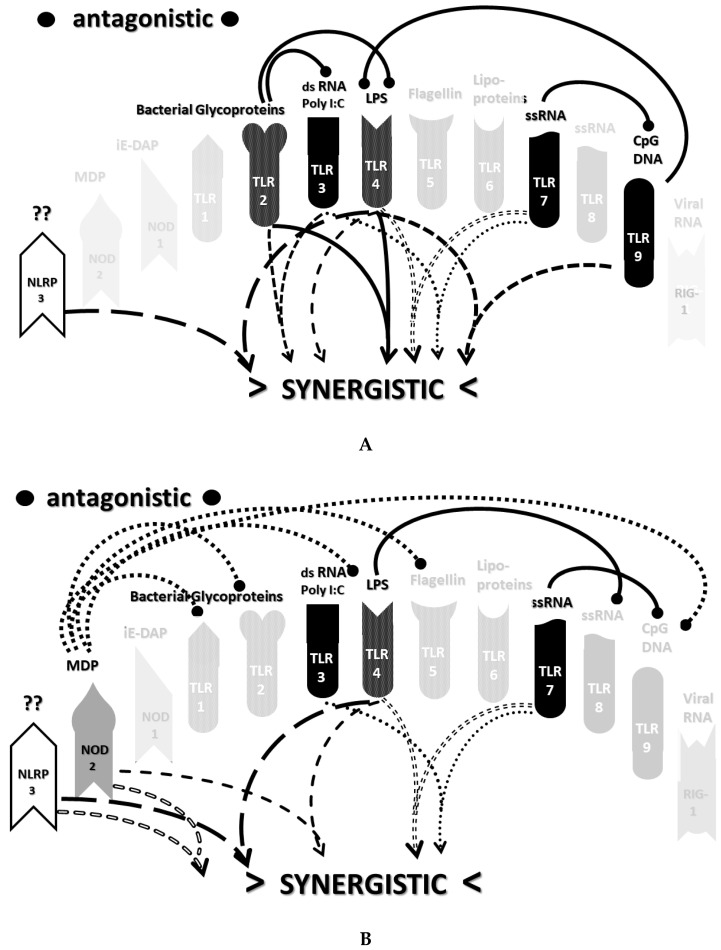
(**A**) Innate receptor activation pattern 2 for severe COVID-19 patients showing synergisms and antagonisms predicted from Table 1 and the model in Figure 2A,B. Receptors that are not known to be activated have been faded. The activation and increased expression of TLR2, TLR3, TLR4, TLR7, TLR9 and NLRP3 (Table 1) results in seven pairs of synergistic cytokine activation pathways. While some of these receptors may also be involved in the antagonisms shown, the fact that the receptors are known to be upregulated and expressed suggests that these antagonisms are either not active or do not outweigh the synergistic effects. The result is multiple synergisms that could drive the cytokine storm associated with severe COVID-19. See Table 1 for abbreviations. (**B**) Innate receptor activation pattern interaction network for influenza-associated ALI/ARDS predicted from Table 1 and the model in Figure 2A,B. TLR3, TLR4, TLR7, NOD2 and NLRP3 are activated, while the other receptors (faded) are not. Note that this is a different set of receptor activations, synergisms and antagonisms that were illustrated for coronaviruses in Figure 3A. The net result is five pairs of receptor synergies with only a single known antagonism on any of the activated receptors (the rest of the antagonisms acting upon receptors that are not upregulated or do not have increased expression in ALI/ARDS). These five synergies may explain how the cytokine storm is driven in ALI/ARDS and, also, why the details of the cytokine storms associated with ALI/ARDS differ in their details from those for sepsis (Figure 3C) or severe COVID-19 (Figure 3A). See Table 1 for abbreviations. (**C**) Innate receptor activation pattern interaction network for sepsis patients predicted from Table 1 and the model in Figure 2A,B. TLR2, TLR4, TLR5, TLR7 and NLRP3 are activated, while the other receptors (faded) are not. Note that this is a different set of receptor activations, synergisms and antagonisms from those illustrated for coronaviruses in Figure 3A or for the influenza A virus in Figure 3B. The net result is five pairs of receptor synergies with only a single known antagonism on any of the activated receptors (the rest of the antagonisms acting upon receptors that are not upregulated or do not have increased expression in sepsis). These five synergies may explain how the cytokine storm is driven in sepsis and, also, why the details of the cytokine storms associated with sepsis differ in their details from those for influenza-associated ALI/ARDS (Figure 3B) or severe COVID-19 (Figure 3A). See Table 1 for abbreviations.

**Figure 4 ijms-22-02108-f004:**
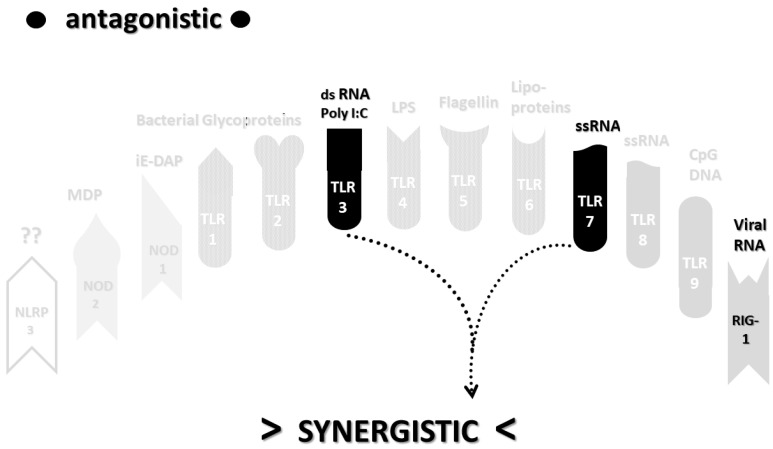
Innate receptor activation pattern for rhinoviruses (from Table 2) based on the model in Figure 2A,B and data summarized in Table 2. Only TLR3, TLR7 and possibly RIG1 are activated, while the other receptors are not (faded), resulting in a single synergistic interaction and no known antagonistic ones. Cytokine release should, according to this model, be minimal following rhinovirus infection, perhaps explaining its mild symptoms. Note the differences in this activation pattern compared with Figure 3A–C and Figure 5 and Figure 6 below). See Table 1 and Table 2 for abbreviations.

**Figure 5 ijms-22-02108-f005:**
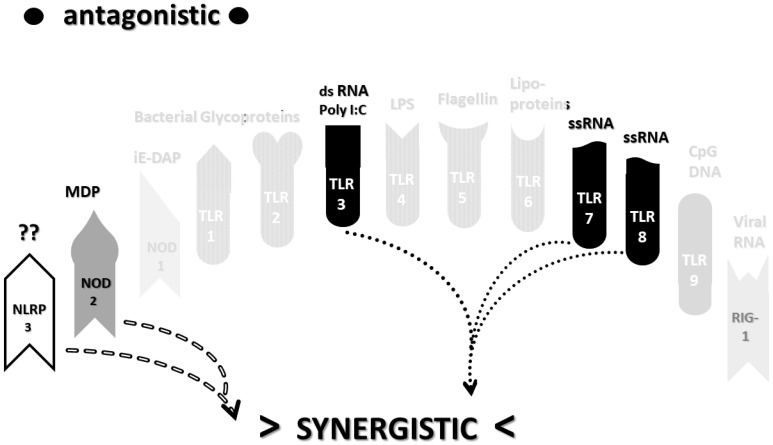
Consensus innate activation profile for coronavirus PAMPs based on the model in Figure 2A,B and data summarized in Table 2. TLR3, TLR7, TLR8, NOD2 and NLRP3 are activated, while the other receptors (faded) are not. The result is two pairs of synergistic interactions (arrows) be Table 3. and TLR7/8 and between NOD2 and NLRP3. No known antagonistic interactions are activated. Cytokine activation should be greater from coronaviruses than from rhinoviruses (Figure 4) but not significantly so, perhaps explaining why most coronavirus infections are asymptomatic or mildly symptomatic and associated with a cold or mild flu symptoms. See Table 1 and Table 2 for abbreviations.

**Figure 6 ijms-22-02108-f006:**
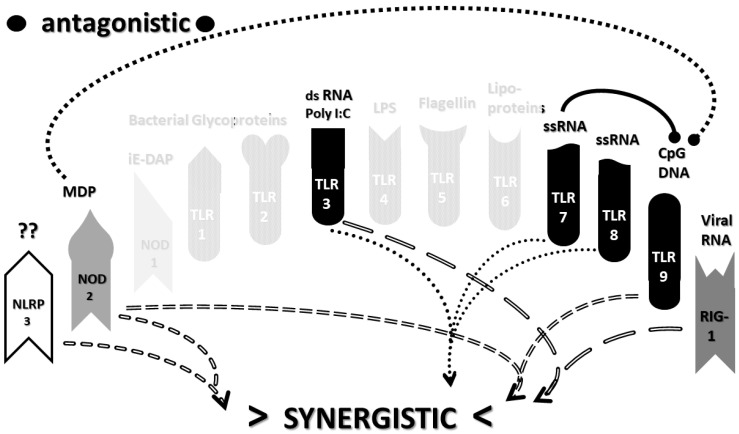
Innate activation patter for respiratory viruses such as influenza A virus PAMPs based on the model in Figure 2A,B and data summarized in Table 2. TLR3, TLR7, TLR8, TLR9, RIG1, NOD2 and NLRP3 are all activated, while the other receptors (faded) are not. Note that the antagonism of NOD2 and TLR7 on TLR9 probably outweighs the synergism between NOD2 and TLR9 so that this synergism is not actually observed. Thus, according to this model, the influenza A virus and other respiratory viruses are likely to induce a greater amount of cytokine releases than rhinovirus (Figure 4) or coronavirus (Figure 5) infections, perhaps explaining why influenza often presents with fevers, chills, joint pain and/or muscle aches—all results of increases in cytokine release—while rhinovirus and coronavirus infections often do not. See Table 1 and Table 2 for abbreviations.

**Figure 7 ijms-22-02108-f007:**
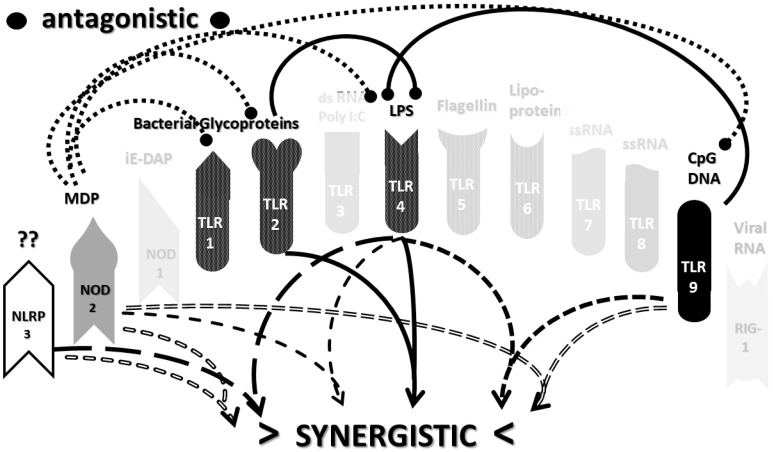
Consensus bacterial innate receptor activation network figure, which also applies well to Aspergillus fungal infections.

**Figure 8 ijms-22-02108-f008:**
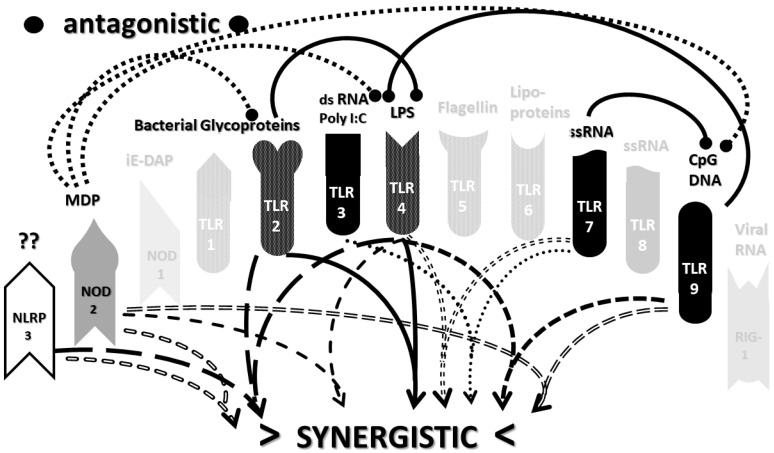
The innate receptor synergisms and antagonisms resulting from the addition of SARS-CoV-2 and bacterial consensus activation patterns (from Table 1 and Table 3). The two antagonistic actions on TLR9 may account for the uncertainty about whether TLR9 is activated or not, despite the coronavirus activation of the receptor. For other TLR, the number of synergisms out-number the antagonisms so that these receptors will presumably continue to be activated. Note the similarity to Figure 3A (the activation pattern in severe COVID-19), the one major difference being the activation of NOD2 in this figure. This activation pattern can vary depending on whether the coinfecting bacterium is Gram-positive, Gram-negative (also activating NOD1) or a mycoplasma (activating neither NOD1 nor NOD2). A very similar pattern would also result from a virus–fungus coinfection such as SARS-CoV-2 as with fungi and yeast coinfections.

**Figure 9 ijms-22-02108-f009:**
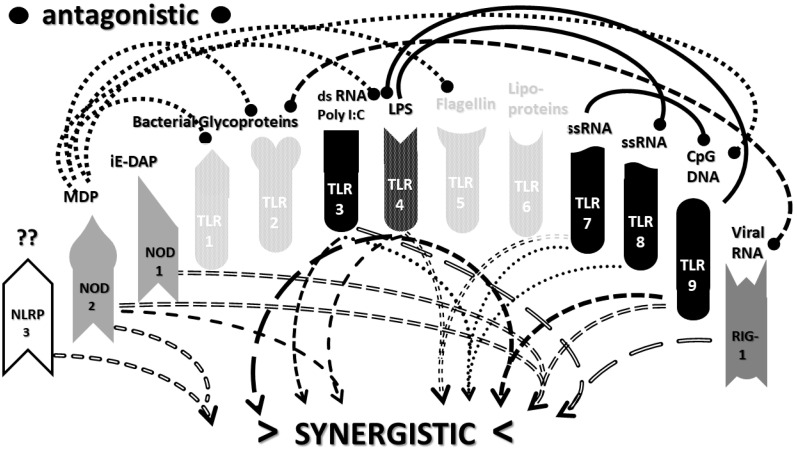
Innate receptor activation pattern resulting from a coinfection of the Influenza A virus and *Haemophilus influenzae* (as often occurred during the Great Influenza Pandemic of 1918–1919) [22,23]. Note the differences from Figure 3A and Figure 8 (severe COVID-19 patients). Note also that there is an antagonistic effect of NOD2 activation on RIG1 and TLR9, an antagonism of TLR7 for TLR9 and an antagonism of TLR4 on TLR8 that may explain the absence of RIG1, TLR8 and TLR9 activation in influenza-associated ALI/ARDS (Table 1), despite the activation of RIG1, TLR8 and TLR9 by the influenza A virus (Table 2). It seems likely that other RIG1 and TLR8 antagonisms (e.g., from NOD1) also exist that are yet to be described. For other TLR, the number of synergisms outnumber the antagonisms, so these receptors will presumably continue to be activated.

**Figure 10 ijms-22-02108-f010:**
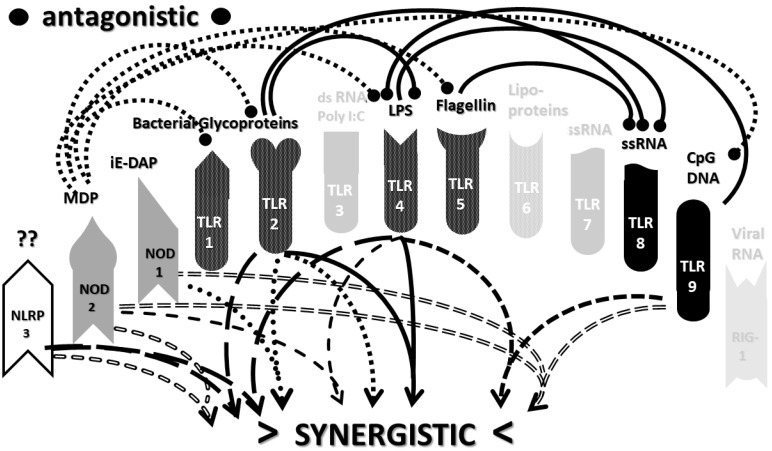
Activation pattern model for polymicrobial sepsis based on a combination of Gram-positive and Gram-negative bacteria from Table 2. Note the several antagonistic interactions on TLR8, which may explain why it is not activated during sepsis. TLR1 is also antagonized (in this case, by NOD2 and, probably, NOD1 as well) and has no synergisms to offset these, so it, too, does not appear to be activated during sepsis (Table 1 and Table 5). For other TLR, synergisms outnumber the antagonisms, so these receptors will presumably continue to be activated with TLR8-activating PAMPs (Table 1 and Table 5). A very similar pattern results from combinations of bacteria with fungi, as can be seen with reference to Table 2 and Table 3. Note the distinct differences from Figure 3A and Figure 8 (severe COVID-19) and Figure 3B and Figure 9 (influenza-associated ALI/ARDS), illustrating the fact that cytokine storms may have distinctly different patterns of innate receptor activation.

**Figure 11 ijms-22-02108-f011:**
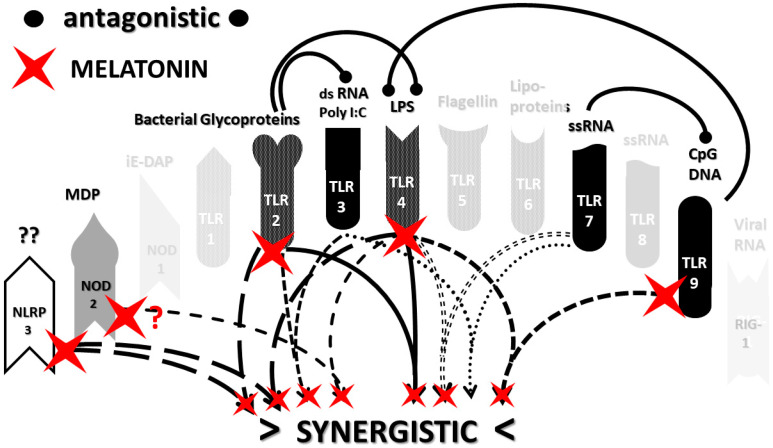
Illustration of the antagonistic effects that melatonin has on TLR and NLR functions in severe COVID-19 (from Figure 3A). The large red stars indicate the TLR and NLR that are antagonized (the question mark indicating that evidence for this antagonism is provisional). The small red stars indicate the synergisms that are eliminated as a result of the receptor antagonisms. By downregulating TLR2, TLR4, TLR9, NLRP3 and, possibly, NOD2 as well, almost all of the innage receptor synergisms driving cytokine overproduction are eliminated, effectively returning the system to normal.

**Figure 12 ijms-22-02108-f012:**
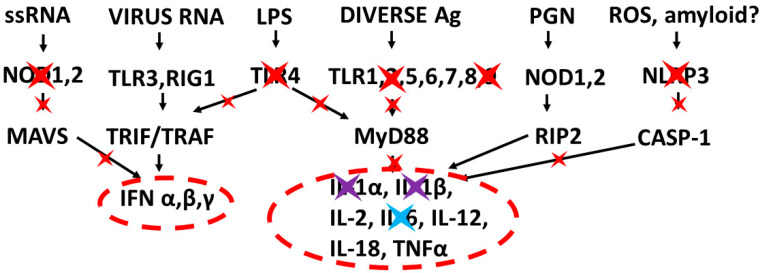
Figure 1 adapted to illustrate the effects of Anakinra, an IL-1 receptor blocker (purple stars), and Tocilizumab and Sarilumab, IL-6 receptor blockers (blue star) that only antagonize one or two cytokines compared with the effects of melatonin (red stars), which downregulates NOD2, TLR2, TLR4, TLR9 and NLRP3, resulting in decreases in the production of all major cytokines (circled in red).

**Table 1 ijms-22-02108-t001:** Summary of studies quantifying the increased activation or protein expression of TLR and NLR for various syndromes associated with cytokine storms. Blank squares indicate that no data were found regarding the activity or expression pattern of the receptor. – indicates no change, ^ indicates increased activation or protein expression, V indicates downregulation or decreased protein expression, +/− indicates conflicting reports about whether there is increased activity or protein expression and -/V indicates conflicting reports about whether there is no change or downregulation. ALI stands for acute lung injury; ARDS stands for acute respiratory distress syndrome; Gram- = expressed by Gram-negative bacteria; PGN = peptidoglycan; LTA = lymphotoxin alpha; HSP = heat shock protein; HMGB1 = High Mobility Group Box 1 protein; dsRNA = double-stranded ribonucleic acid; polyI:C = polymer composed of inosine and cytosine; DAMP = damage-associated proteins; Gram+ = expressed by Gram-positive bacteria; lipopeps = lipopolypeptides; ssRNA = single-stranded ribonucleic acids; CpG DNA = deoxyribonucleic acid high in cytosine and guanine, typifying a microbial genomic origin; mtDNA = deoxyribonucleic acid derived from mitochondria; DAP = diaminopimelic acid, a bacterial cell wall component; MDP = muramyl dipeptide, a bacterial cell wall component and ? = activators not presently known.

Receptors	TLR1	TLR2	TLR3	TLR4	TLR5	TLR6	TLR7	TLR8	TLR9	TLR10	NOD1	NOD2	NL-RP3	RIG1
Activated by:	**Lipo-peptides**	**Gram-PGN, LTA** **& HSP, HMGB1**	**ds-RNA, polyI:C,** **DAMP**	**Gram+** **LPS & HSP, HMGB1**	**Flagellin**	**LTA, lipopeps**	**ss-RNA**	**ss-** **RNA** **& pyogenic Bacteria**	**CpG DNA** **& mtDNA**	**Retroviral RNA**	**Meso-DAP**	**MDP**	**?**	**Viral RNA**
COVID-19 SEVERE	**-**	**^**	**+/-**	**^**	**-**	**-**	**^**	**-**	**+/-**	**-**	**-**	**-**	**^**	**-**
INFLUENZA-ASSOCIATED ALI/ARDS		**v**	**^**	**^**			**^**	**-**	**^**		**+/-**	**^**	**^**	**v**
SEPSIS/ALI (murine)		**^**	**+/-**	**^**	**-**		**^**		**^**		**-**	**-**	**^**	
SEPSIS human patients	**-**	**+/-**	**-**	**^**	**+/-**	**-**	**+/-**	**-**	**-/v**	**-**	**-**	**-**	**^**	
Polymicrobial SEPSIS murine model	**-**	**^**	**-**	**^**	**-**	**-**	**^**	**-**	**-**		**-**	**-**	**^**	
SEPSIS CONCENSUS	**-**	**^**	**-**	**^**	**-**	**-**	**^**	**-**	**-**		**-**	**-**	**^**	

**Table 2 ijms-22-02108-t002:** Summary of studies quantifying the increased activation or protein expression of TLR and NLR for innate immune system cells following viral infections. CoV229E = coronavirus type 229E (a cold virus); SARS-CoV-1 = severe acute respiratory syndrome coronavirus type 1; MERS = Middle East respiratory syndrome virus; SARS-CoV-2 = severe acute respiratory syndrome coronavirus type 2 and RESP = respiratory. Blank squares indicate that no data were found regarding the activity or expression pattern of the receptor. – indicates no change, ^ indicates increased activation or protein expression, V indicates downregulation or decreased protein expression, +/- indicates conflicting reports about whether there is increased activity or protein expression and -/V indicates conflicting reports about whether there is no change or downregulation. PGN = peptidoglycan; LTA = lymphotoxin alpha; HSP = heat shock protein; HMGB1 = High Mobility Group Box 1 protein; dsRNA = double-stranded ribonucleic acid; polyI:C = polymer composed of inosine and cytosine; DAMP = damage-associated proteins; Gram+ = expressed by Gram-positive bacteria; lipopeps = lipopolypeptides; ssRNA = single-stranded ribonucleic acids; CpG DNA = deoxyribonucleic acid high in cytosine and guanine, typifying a microbial genomic origin; mtDNA = deoxyribonucleic acid derived from mitochondria; DAP = diaminopimelic acid, a bacterial cell wall component; MDP = muramyl dipeptide, a bacterial cell wall component and ? = activators not presently known.

Receptor:	TLR 1	TLR 2	TLR 3	TLR 4	TLR 5	TLR 6	TLR 7	TLR 8	TLR 9	TLR 10	NOD 1	NOD 2	NL-RP 3	RIG 1
Activated by:	**Lipopeptides**	**Gram- PGN, LTA** **& HSP, HMGB1**	**ds-RNA, polyI:C,** **DAMP**	**Gram+** **LPS & HSP, HMGB1**	**Flagellin**	**LTA, lipopeps**	**ss-RNA**	**ss-** **RNA** **& pyogenic Bacteria**	**CpG DNA** **& mtDNA**	**Retroviral RNA**	**Meso-DAP**	**MDP**	**?**	**Viral RNA**
CoV 229E		**-**		**-**	**-**		**^**		**-**					v
SARS-CoV-1	**-**	**+/-**	**^**	**-**	**-**	**-**	**^**	**^**	**v**	**-**			**^**	v
MERS			**^**	**v**			**^**					**+/-**	**^**	v
SARS-CoV-2		**+/-**	**^**				**^**					**^**	**^**	v
Coronavirus CONCENSUS			**^**				**^**						**^**	v
Influenza A viruses	**-**	**v**	**^**	**v**	**-**	**-**	**^**	**^**	**^**	**-**		**^**	**^**	**^**
Rhinoviruses			**^**				**^**	**-**						**+/-**
Respiratory syncytial virus	**-**	**-**	**^**	**+/-**		**-**	**^**	**^**	**+/-**			**^**	**^**	**^**
Adenovirus			**^**				**^**		**^**				**^**	**^**
Coxsackie-viruses			**^**				**^**	**^**	**^**			**^**	**^**	**^**
RESP VIRUS CONCENSUS	**-**	**-**	**^**	**-**	**-**	**-**	**^**	**^**	**^**			**^**	**^**	**^**

**Table 3 ijms-22-02108-t003:** Summary of studies quantifying the increased activation or protein expression of TLR and NLR for innate immune system cells following bacterial infections associated with severe COVID-19, ALI/ARDS and/or sepsis. Blank squares indicate that no data were found regarding the activity or expression pattern of the receptor. GRAM POS = Gram-positive bacterium, GRAM AMBI = bacterium that yields ambiguous Gram staining, GRAM NEG = Gram-negative bacterium and NO GRAM means that Gram staining is irrelevant. – indicates no change, ^ indicates increased activation or protein expression, V indicates downregulation or decreased protein expression, +/- indicates conflicting reports about whether there is increased activity or protein expression, -/V indicates conflicting reports about whether there is no change or downregulation and a blank square indicates no available information. PGN = peptidoglycan; LTA = lymphotoxin alpha; HSP = heat shock protein; HMGB1 = High Mobility Group Box 1 protein; dsRNA = double-stranded ribonucleic acid; polyI:C = polymer composed of inosine and cytosine; DAMP = damage-associated proteins; Gram+ = expressed by Gram-positive bacteria; lipopeps = lipopolypeptides; ssRNA = single-stranded ribonucleic acids; CpG DNA = deoxyribonucleic acid high in cytosine and guanine, typifying a microbial genomic origin; mtDNA = deoxyribonucleic acid derived from mitochondria; DAP = diaminopimelic acid, a bacterial cell wall component; MDP = muramyl dipeptide, a bacterial cell wall component and ? = activators not presently known.

	TLR 1	TLR 2	TLR 3	TLR 4	TLR 5	TLR 6	TLR 7	TLR 8	TLR 9	TLR 10	NOD 1	NOD 2	NL-RP3	RIG 1
	Lipopeptides	Gram- PGN, LTA & HSP, HMGB1	ds-RNA, polyI:C, DAMP	Gram+ LPS & HSP, HMGB1	Flagellin	LTA, lipopeps	ss-RNA	ss- RNA & pyogenic Bacteria	CpG DNA & mtDNA	Retroviral RNA	Meso-DAP	MDP	?	Viral RNA
**GRAM POS**														
*Group A Streptococci*	**^**	**^**	**-**	**^**	**-**	**-**		**-**	**^**	**-**	**-**	**^**	**^**	
*Group B Streptococci*		**-**					**^**	**^**	**^**		**-**	**-**	**^**	
*Staphylococcus aureus*		**^**		**^**				**^**	**^**		**-**	**^**	**^**	
**GRAM AMBI**														
*Mycobacterium tuberculosis*	**^**	**^**	**+/-**	**^**					**^**		**^**	**^**		
**GRAM NEG**														
*Klebsiella pneumoniae*		**^**		**^**					**^**		**^**		**^**	
*Haemophilus influenzae*		**^**		**+/-**					**-**		**^**	**^**	**^**	
*Legionella pneumophila*		**^**			**^**				**^**		**-**	**-**	**^**	
*Chlamydia pneumoniae*		**^**		**^**							**^**	**^**	**^**	
*Neisseria meningitidis*	**^**	**^**		**^**					**^**		**^**		**^**	
*Pseudomonas aeruginosa*				**^**	**^**				**^**		**^**	**^**	**^**	
**NO GRAM**														
*Mycoplasma pneumoniae*	**^**	**^**		**^**		**^**					**-**	**-**	**^**	
**BACTERIA CONCENSUS**	**^**	**^**		**^**					**^**		**+/-**	**^**	**^**	

**Table 4 ijms-22-02108-t004:** Innate immune receptor activation by fungi associated with severe COVID-19. Spp. = species, – indicates no change, ^ indicates increased activation or protein expression, V indicates downregulation or decreased protein expression, +/- indicates conflicting reports about whether there is increased activity or protein expression and a blank square indicated no available information. PGN = peptidoglycan; LTA = lymphotoxin alpha; HSP = heat shock protein; HMGB1 = High Mobility Group Box 1 protein; dsRNA = double-stranded ribonucleic acid; polyI:C = polymer composed of inosine and cytosine; DAMP = damage-associated proteins; Gram+ = expressed by Gram-positive bacteria; lipopeps = lipopolypeptides; ssRNA = single-stranded ribonucleic acids; CpG DNA = deoxyribonucleic acid high in cytosine and guanine, typifying a microbial genomic origin; mtDNA = deoxyribonucleic acid derived from mitochondria; DAP = diaminopimelic acid, a bacterial cell wall component; MDP = muramyl dipeptide, a bacterial cell wall component and ? = activators not presently known.

	TLR 1	TLR 2	TLR 3	TLR 4	TLR 5	TLR 6	TLR 7	TLR 8	TLR 9	TLR 10	NOD 1	NOD 2	NL-RP 3	RIG 1
	Lipopeptides	Gram- PGN, LTA & HSP, HMGB1	ds-RNA, polyI:C, DAMP	Gram+ LPS & HSP, HMGB1	Flagellin	LTA, lipopeps	ss-RNA	ss-RNA & pyogenic Bacteria	CpG DNA & mtDNA	Retroviral RNA	Meso-DAP	MDP	?	Viral RNA
*Aspergillus* spp.		**^**	**^**	**^**					**^**		**^**	**^**	**^**	
*Candida* spp.		**^**		**^**			**^**	**^**	**^**			**-**	**^**	
*Cryptococcus* spp.		**^**	**^**	**^**					**^**				**^**	
FUNGI CONSENSUS		**^**	**^**	**^**					**^**				**^**	

**Table 5 ijms-22-02108-t005:** Summary table of the consensus activation patterns of microbes derived from the previous Table 1, Table 2, Table 3 and Table 4. TLR and NLR columns in white are those activated mainly by bacterial and fungal PAMPs; columns in grey are those activated mainly by virus PAMPs. Only NLRP3 (black boxes with white symbols) is activated by all of the pathogens and is upregulated in all of the cytokine storm-related diseases listed here. The G+G- symbol under NOD1 for the bacterial consensus represents the fact that Gram-positive bacteria do not activate NOD1 but Gram-negative bacteria do. – indicates no change, ^ indicates increased activation or protein expression, V indicates downregulation or decreased protein expression, +/- indicates conflicting reports about whether there is increased activity or protein expression, -/V indicates conflicting reports about whether there is no change or downregulation and a blank square indicates no available information. See Table 1 for the rest of the abbreviations.

	TLR1	TLR2	TLR3	TLR4	TLR5	TLR6	TLR7	TLR8	TLR9	TLR10	NOD1	NOD2	NLRP 3	RIG1
@@	**Lipo-peptides**	**Gram- PGN, LTA** **& HSP, HMGB1**	**ds-RNA, polyI:C,** **DAMP**	**Gram+** **LPS & HSP, HMGB1**	**Flagellin**	**LTA, lipopeps**	**ss-RNA**	**ss-** **RNA** **& pyogenic Bacteria**	**CpG DNA** **& mtDNA**	**Retroviral RNA**	**Meso-DAP**	**MDP**	**?**	**Viral RNA**
Coronavirus CONSENSUS			**^**				**^**						**^**	v
RESP VIRUS CONCENSUS	**-**	**-**	**^**	**-**	**-**	**-**	**^**	**^**	**^**			**^**	**^**	**^**
BACTERIA CONSENSUS		**^**		**^**					**^**		**G+ G-**	**^**	**^**	
FUNGI CONSENSUS		**^**	**^**	**^**					**^**				**^**	
COVID-19 SEVERECONSENSUS	**-**	**^**	**+/-**	**^**	**-**	**-**	**^**	**-**	**+/-**	**-**	**-**	**-**	**^**	**-**
INFLUENZA-ASSOCIATED ALI/ARDS		**v**	**^**	**^**			**^**	**-**	**^**		**+/-**	**^**	**^**	v
SEPSIS human patients	**-**	**+/-**	**-**	**^**	**+/-**	**-**	**+/-**	**-**	**-**/v	**-**	**-**	**-**	**^**

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
