# Peer review of "Innate Receptor Activation Patterns Involving TLR and NLR Synergisms in COVID-19, ALI/ARDS and Sepsis Cytokine Storms: A Review and Model Making Novel Predictions and Therapeutic Suggestions"

_ijms, 2021, doi:10.3390/ijms22042108_

Round 1
Reviewer 1 Report
Dear Authors,
I have read the manuscript with interest and some questions raised. Attached please find my comments.
Overall. General English grammar revision (Minor spelling errors).
Abstract. Provide a structured summary including, as applicable: background; objectives; data sources; study eligibility criteria; study appraisal and synthesis methods; results; conclusions and implications of key findings.
Introduction. Authors stated “Some patients experiencing severe COVID-19, the disease caused by the SARS-CoV-2 38 betacoronavirus, develop what is sometimes described as a “cytokine storm” or “cytokine release syndrome” characterized by over-stimulation of macrophages, dendritic cells and monocytes producing the cytokines interleukin 1 (IL1), interleukin 6 (IL6), interleukin 10 (IL10), tumor necrosis factor alpha (TNFα), tumor necrosis factor beta (TNFβ), and ferritin”. Please add a reference for this statement.
Introduction. Authors stated “These cytokines produce eosinopenia and lymphocytopenia characterized by low counts of eosinophils, CD8+ T cells, natural killer (NK) and naïve T helper cells, simultaneously inducing naive B-cell activation, increased Th17 lymphocyte differentiation, and stimulation of monocyte and neutrophil recruitment”. Please add a reference for this statement.
Introduction. Authors stated “The problem of what causes “cytokine storms” or “cytokine release syndromes” extends beyond COVID-19 more generally to sepsis, acute lung injury (ALI) and acute respiratory distress syndrome (ARDS) associated with other respiratory infections”. Please add a reference for this statement.
Please add a section explaining the methods used to perform literature review.
State the process for selecting studies (i.e., screening, eligibility, included in review).
Give numbers of studies screened, assessed for eligibility, and included in the review, with reasons for exclusions at each stage, ideally with a flow diagram.
Please add a table showing the risk of bias with a quality assessment of the main studies taken into account to perform revision. Please use the three codified colors to identify low (green), moderate (yellow) or high (red) risk of bias. Please add details about how this information is to be used in any data synthesis.
Discussion. Please remove “?” from the title.
Discussion. Discuss limitations at study and outcome level (e.g., risk of bias), and at review-level (e.g., incomplete retrieval of identified research, reporting bias).
Discussion. Provide a specific interpretation of the results with particular reference to the possible implications for future research.
Figures: ok
Table 1. Please move the captions of the table on the top of the table, and possibly in the same page of the table itself.
Table 2. Please move the captions of the table on the top of the table, and possibly in the same page of the table itself.
Tables: Please add an additional table concerning the risk of bias.
References. Huge list. I wonder if the reference list could be reduced, (if possible).
Author Response
I have read the manuscript with interest and some questions raised. Attached please find my comments.
Overall. General English grammar revision (Minor spelling errors)
SPELLING ERRORS HAVE BEEN CORRECTED.
Abstract. Provide a structured summary including, as applicable: background; objectives; data sources; study eligibility criteria; study appraisal and synthesis methods; results; conclusions and implications of key findings.
THIS FORMAT IS NOT REQUIRED BY THIS JOURNAL NOR IS IT APPROPRIATE FOR THE TYPE OF REVIEW-HYPOTHESIS PAPER I HAVE WRITTEN. SEE BELOW FOR THE METHODOLOGICAL REASIONS THAT DATA SOURCES, ELIGIBILITY CRITERIA, STUDY APPRAISAL METHODS, ETC. ARE NOT ONLY IRRELEVANT BUT UNAVAILABLE FOR THE TYPES OF SOURCES USED HERE.
Introduction. Authors stated “Some patients experiencing severe COVID-19, the disease caused by the SARS-CoV-2 38 betacoronavirus, develop what is sometimes described as a “cytokine storm” or “cytokine release syndrome” characterized by over-stimulation of macrophages, dendritic cells and monocytes producing the cytokines interleukin 1 (IL1), interleukin 6 (IL6), interleukin 10 (IL10), tumor necrosis factor alpha (TNFα), tumor necrosis factor beta (TNFβ), and ferritin”. Please add a reference for this statement.
ADDED
Introduction. Authors stated “These cytokines produce eosinopenia and lymphocytopenia characterized by low counts of eosinophils, CD8+ T cells, natural killer (NK) and naïve T helper cells, simultaneously inducing naive B-cell activation, increased Th17 lymphocyte differentiation, and stimulation of monocyte and neutrophil recruitment”. Please add a reference for this statement.
ADDED
Introduction. Authors stated “The problem of what causes “cytokine storms” or “cytokine release syndromes” extends beyond COVID-19 more generally to sepsis, acute lung injury (ALI) and acute respiratory distress syndrome (ARDS) associated with other respiratory infections”. Please add a reference for this statement.
ADDED
Please add a section explaining the methods used to perform literature review.
DONE: NEW SECTION 2.5 ADDED
State the process for selecting studies (i.e., screening, eligibility, included in review).
THE ALTERNATIVE HYPOTHESES TO BE TESTED DETERMINE WHAT STUDIES ARE APPROPRIATE FOR TESTING THEM. (THIS IS NOT A META-STUDY TRYING TO INTEGRATE A BUNCH OF CLINICAL TRIALS AND THAT MODEL IS COMPLETELY INAPPROPRIATE FOR THIS PAPER!) THE CRITERION IS WHETHER A PAPER PROVIDES DATA THAT TEST SOME ASPECT OF THE HYPOTHESIS.
Give numbers of studies screened, assessed for eligibility, and included in the review, with reasons for exclusions at each stage, ideally with a flow diagram.
NOT RELEVANT. THIS REQUEST ASSUMES A MODEL OF HOW REVIEWS SHOULD BE WRITTEN THAT IS BASED ON CLINICAL STUDIES THAT IS COMPLETELY INAPPROPRIATE TO THE NATURE OF THE INFORMATION BEING USED, AND IRRELEVANT TO TESTING AN HYPOTHESIS ABOUT HOW A SYSTEM WORKS. AGAIN, THIS IS NOT A METASTUDY IN WHICH PAPERS ARE EVALUATED FOR STATISTICAL VALIDITY, POPULATION SIZE, DOUBLE-BLINDING, SINGLE-BLINDING, ETC. – THE VAST MAJORITY OF THE STUDIES CITED IN THIS PAPER ARE LABORATORY STUDIES THAT DO NOT INCLUDE STATISTICAL EVALUATIONS NOR “POPULATION SIZES” NOR BLINDING OF ANY KIND. VALIDITY IS DETERMINED IN BENCH SCIENCE BY THE METHODOLOGY USED, REPRODUCIBILITIY AND EXTENDABILITY TO OTHER TISSUES OR ORGANISMS, NOT THE SIZE OF THE STUDY POPULATION, ETC. SUCH ISSUES ARE NOT AMENABLE TO TABLULATION, FLOW DIAGRAMS OR ANY OF THE OTHER EVALUATION METHODS USED IN METASTUDIES OF CLINICAL STUDIES.
Please add a table showing the risk of bias with a quality assessment of the main studies taken into account to perform revision. Please use the three codified colors to identify low (green), moderate (yellow) or high (red) risk of bias. Please add details about how this information is to be used in any data synthesis.
AGAIN, COMPLETELY INAPPROPRIATE TO THE TYPES OF STUDIES REVIEWED AND IRRELVANT AND IMPOSSIBLE GIVEN THEIR METHODS.
Discussion. Please remove “?” from the title.
DONE
Discussion. Discuss limitations at study and outcome level (e.g., risk of bias), and at review-level (e.g., incomplete retrieval of identified research, reporting bias).
ADDED: SECTION 3.5
Discussion. Provide a specific interpretation of the results with particular reference to the possible implications for future research.
ADDED: SECTION 3.4
Figures: ok
Table 1. Please move the captions of the table on the top of the table, and possibly in the same page of the table itself.
DONE
Table 2. Please move the captions of the table on the top of the table, and possibly in the same page of the table itself.
DONE
ALSO DONE FOR THE OTHER TABLES, BY THE WAY!
Tables: Please add an additional table concerning the risk of bias.
THIS IS NOT POSSIBLE FOR THE REASONS DISCUSSED ABOVE CONCERING METHODS. THESE ARE NOT CLINICAL STUDIES THAT HAVE MEASURES OF STATISTICAL VALIDITY, POPULATION SIZES, ETC. THESE ARE MAINLY IN VITRO STUDIES. THERE IS THEREFORE NO WAY TO TABULATE RISK OF BIAS.
IN FACT, WHAT IS PRESENTED IS AN HYPOTHESIS LINKING THE DATA INTO A COHERENT STORY. AS SUCH, THE HYPOTHESIS IS INTRINSICALLY BIASED IN THE SAME WAY EVERY HYPOTHESIS IS BIASED: IT PROPOSES THAT CERTAIN FACTORS ARE IMPORTANT AND OTHERS ARE NOT. THE INHERENT BIAS IS CONTROLLED FOR BY COMPARING THE HYPOTHESIS AGAINST ALTERNATAIVE HYPOTHESES. THE POINT OF PROPOSING A HYPOTHESIS IS TO MAKE SENSE OF PREVIOUSLY UNRELATED OR PROBLEMATIC FINDINGS (SUCH AS WHY CYTOKINE STORMS SEEM TO COME IN SO MANY DIFFERENT VERSIONS) AND TO COMPARE THE EXTENT TO WHICH THE HYPOTHESIS MAKES SENSE OF THE PROBLEMS IN WAYS THAT ALTERNATIVE HYPOTHESES DO NOT. THE CRITERIA FOR DOING THIS ARE AS OLD AS SCIENCE: DO THE DATA FIT PREDICTIONS; DO ANOMALOUS DATA EXIST THAT CANNOT BE EXPLAINED; IS THE EXPLANATION LOGICAL AND COHERENT; ETC. FINALLY, A NOVEL HYPOTHESIS SHOULD MAKE TESTABLE PREDICTIONS THAT OTHER HYPOTHESES DO NOT. I HAVE DONE ALL OF THESE THINGS IN THIS PAPER. THE QUESTION IS NOT WHETHER THE RESULT IS BIASED (OF COURSE IT IS: IT’S MY CURRENTLY UNIQUE VIEW OF THIS FIELD!), BUT WHETHER IT PROVIDES ACTIONABLE INSIGHTS AND USEFUL OPPORTUNITIES FOR OTHER INVESTIGATORS AND CLINICIANS THAT ARE MISSING FROM OUR CURRENT UNDRESTANDING OF THE ISSUES AND THAT DO NOT PROCEED FROM ALTERNATIVE HYPOTHESES.
IF YOU NEED FURTHER JUSTIFICATION FOR THIS APPROACH, SEE MY BOOK, DISCOVERING: FINDING AND SOLVING PROBLEMS AT THE FRONTIERS OF KNOWLEDGE (HARVARD UNIVERSITY PRESS, 1989).
References. Huge list. I wonder if the reference list could be reduced, (if possible).
IT IS NOT POSSIBLE. I HAVE ALREADY USED THE MINIMUM NUMBER OF REFERENCES TO MAKE EACH KEY POINT. THIS MANUSCRIPT INTEGRATES STUDIES FROM EIGHT DIFFERENT SPECIALTIES, MOST OF WHICH DO NOT HAVE INTEGRATED REVIEWS EVEN WITHIN THAT SPECIALTY: 1) INNATE RECEPTOR SYNERGIES AND ANTAGONISMS; 2) SPECIFIC ACTIVATION OF INNATE RECEPTORS BY INDIVIDUAL MICROBES ASSOCIATED WITH COVID-19; 3) INNATE RECEPTOR ACTIVATION IN COVID-19; 4) SPECIFIC ACTIVATION OF INNATE RECEPTORS BY INDIVIDUAL MICROBES ASSOCIATED WITH ALI/ARDS; 5) INNATE RECEPTOR ACTIVATION IN ALI/ARDS; 6) SPECIFIC ACTIVATION OF INNATE RECEPTORS IN SEPSIS; 7) INNATE ACTIVATION IN SEPSIS; 8) INNATE RECEPTOR ANTAGONISM BY VARIOUS TREATMENTS FOR CYTOKINE STORMS. THAT’S EIGHT SETS OF LITERATURE, AVERAGING ONLY ABOUT 30 ARTICLES PER SET – HARDLY EXCESSIVE!
Submission Date
27 January 2021
Date of this review
03 Feb 2021 11:45:43
Bottom of Form
© 1996-2021 MDPI (Basel, Switzerland) unless otherwise stated
Disclaim
Top of Form
Reviewer 2 Report
In this review article, the authors discussed the unique clinical, experimental and therapeutic predictions, and broader implications are outlined for understanding why other syndromes such as acute lung injury, acute respiratory distress syndrome and sepsis display varied cytokine storm symptoms.
Comments
This is an interesting review article. This manuscript is well-written. The reviewer has only some minor concerns as follows:
- Figures 2-11, especially Figure 2, are complex and are easy to confuse. The different types of lines may need to clearly describe in detail. The different colors may be considered.
- The position of Table 3 can be replaced. Moreover, the symbol for “@@” in the second row heading should be described.
Author Response
Comments
This is an interesting review article. This manuscript is well-written. The reviewer has only some minor concerns as follows:
- Figures 2-11, especially Figure 2, are complex and are easy to confuse. The different types of lines may need to clearly describe in detail. The different colors may be considered.
I AGREE ABOUT FIGURE 2. I HAVE THEREFORE BROKEN IT INTO TWO, SLIGHTLY LESS CONFUSING FIGURES (NOW 2A and 2B) SEPARATING OUT THE TLR-TLR INTERACTIONS FROM THE TLR-NLR AND NLR-NLR INTERACTIONS (THERE ARE TOO FEW NLR-NLR INTERACTIONS TO JUSTIFY A SEPARATE FIGURE FOR THEM!).
THE PURPOSE OF THE DIFFERENT TYPES OF LINES WAS INTENDED SIMPLY TO MAKE IT EASIER TO FOLLOW WHICH PAIRS OF TLR/NLR ARE INTERACTING AND HAVE NO FURTHER SIGNIFICANCE. I HAVE NOW CLARIFIED THAT POINT IN THE FIGURE CAPTIONS. I CHOSE (AND CONTINUE TO CHOOSE) NOT TO USE COLORED LINES BECAUSE I FOUND THAT THE MULTIPLE COLORS ACTUALLY ADD TO THE VISUAL CONFUSION! (IMAGINE FIGURE 2 WITH AN ENTIRE RAINBOW OF COLORED LINES – PRETTIER, BUT CERTAINLY NOT CLEARER!)
- The position of Table 3 can be replaced. Moreover, the symbol for “@@” in the second row heading should be described.
THE TABLE POSITION IS MOVED (THOUGH THIS WILL PROBABLY CHANGE AGAIN WHEN THE ARTICLE IS TYPE SET!) AND THE “@@” HAS BEEN REMOVED AS IT WAS AN ARTIFACT OF AN EARLIER VERSION OF THE TABLE AND HAS NO MEANING.
Round 2
Reviewer 1 Report
Good job